# Developing a novel force forecasting technique for early prediction of critical events in robotics

**Meenakshi Narayan**[1]*, **Ann Majewicz Fey**[1,2]

**1** Department of Mechanical Engineering, The University of Texas at Dallas, Richardson, Texas, United States of America, **2** Department of Surgery, UT Southwestern Medical Center, Dallas, Texas, United States of America

* Meenakshi.Narayan@utdallas.edu

**Data Availability Statement:** Replication data for "Developing a novel force forecasting technique for early prediction of critical events in robotics" are available from the Harvard Dataverse at https://doi.org/10.7910/DVN/R7ZLHZ.

## Abstract

Safety critical events in robotic applications can often be characterized by forces between the robot end-effector and the environment. One application in which safe interaction between the robot and environment is critical is in the area of medical robots. In this paper, we propose a novel Compact Form Dynamic Linearization Model-Free Prediction (CFDL-MFP) technique to predict future values of any time-series sensor data, such as interaction forces. Existing time series forecasting methods have high computational times which motivates the development of a novel technique. Using Autoregressive Integrated Moving Average (ARIMA) forecasting as benchmark, the performance of the proposed model was evaluated in terms of accuracy, computation efficiency, and stability on various force profiles. The proposed algorithm was **11%** more accurate than ARIMA and maximum computation time of CFDL-MFP was **4ms**, compared to ARIMA (7390ms). Furthermore, we evaluate the model in the special case of predicting needle buckling events, before they occur, by using only axial force and needle-tip position data. The model was evaluated experimentally for robustness with steerable needle insertions into different tissues including gelatin and biological tissue. For a needle insertion velocity of 2.5mm/s, the proposed algorithm was able to predict needle buckling **2.03s** sooner than human detections. In biological tissue, no false positive or false negative buckling detections occurred and the rates were low in artificial tissue. The proposed forecasting model can be used to ensure safe robot interactions with delicate environments by predicting adverse force-based events before they occur.

## 1 Introduction

The technique of predicting future events based on past and present information of a system has played a significant role in framing optimal and safe decisions for various applications such as mobile surveillance, high-performance manipulation and medical applications to name a few [1]. Accuracy and timely predictions of a system state in response to environmental conditions enhance the ability to provide appropriate control decisions for safe system operations. However, given the unknown dynamic structure of environments, it is challenging to control the system's response to its environment. Through sensing, estimation and

**Funding:** Research reported in this publication was supported by the National Center for Advancing Translational Sciences of the National Institutes of Health under award Number UL1TR001105. The content is solely the responsibility of the authors and does not necessarily represent the official views of the NIH.

**Competing interests:** The authors have declared that no competing interests exist.

prediction, information of the system state and environment can be approximately retrieved [1]. There has been growing interest in developing prediction-based control strategies through sensors, to prevent undesirable events in autonomous intelligent robots, human-robot interactions, and multi-agent robots, using some danger metric. This approach of using metrics to foresee an upcoming danger enables the robot to take preventive control actions before the danger occurs [2–4].

One critically important area for adverse event prediction is in the field of medicine. In the areas of medical decision making, time series forecasting models have been successfully applied to predict mortality and time dependent risks [5], early diagnosis of disease [6], and heartbeat rates to estimate activity-based physiological response [7]. However, there is a paucity of research related to prediction-based control for medical robotic systems, and in particular, surgical robots.

## 1.1 Motivation

In surgery, patient safety has been defined in terms of how quickly the surgeon responds to critical adverse events and performs appropriate control actions in real time to minimize risks to the patient [8]. Adverse events including tool malfunctions, operator errors, and improper control of surgical robots have been shown to cause negative patient outcomes, such as injuries and deaths. For example, during the past decade, tool malfunctions alone constituted 62% of injury related events, with imaging errors and broken tool-tips contributing to 7.4% and 10.5% of adverse events [9]. Recently, efforts have been made to facilitate turn-taking prediction in surgeon-robot collaboration for early delivery of surgical instruments, thereby enhancing safe and efficient procedure [10]. Their work is related to human-robot collaboration; however, there is still an important and unaddressed area of research related to predicting adverse environmental conditions. In this paper, we will focus on predicting critical events that are common in robotic needle-based interventions.

In robotic needle steering, flexible asymmetric-tip needles are designed to automatically steer within a tissue to reach specific target locations. This technology has tremendous potential to assist surgeons in performing many percutaneous procedures and needle-based interventions with enhanced precision, dexterity, and reliability [11–14]. However, critical issues related to patient safety and robust system performance have limited the clinical applicability of robotically steered needles [15]. The highly dynamic and unpredictable nature of steerable needles make them a perfect case study for developing a novel force prediction technique, with force being the time series data measured from a force sensor.

Forces exerted from the surrounding tissue can increase rapidly causing the needle to laterally bend and buckle when encountering stiff tissue [16, 17]. On further insertion, either the needle tip could break, or membrane puncture can occur causing undesired tissue rupture and bleeding [18]. Therefore, buckling can cause damage to both the needle and the tissue [19, 20]. In order to lower the risks of this damage, it is crucial to minimize the lag between the time of event occurrence and the time of event detection. We attempted to minimize this lag through our previously developed algorithm that detected needle buckling events within 2mm after encounter with a hard impenetrable obstacle, and at least one second sooner than human detection [21]. However, detecting an adverse event on occurrence can still be too late to prevent any potential damage to the needle-tissue system [8]. Thus, it is more crucial to predict a buckling event before it occurs so that an appropriate decision can be implemented to prevent the needle from buckling, as illustrated in Fig 1. Since nearly constant needle-tip position and force patterns were used to characterize needle buckling [21], the first step is to perform forecasts of axial force data.

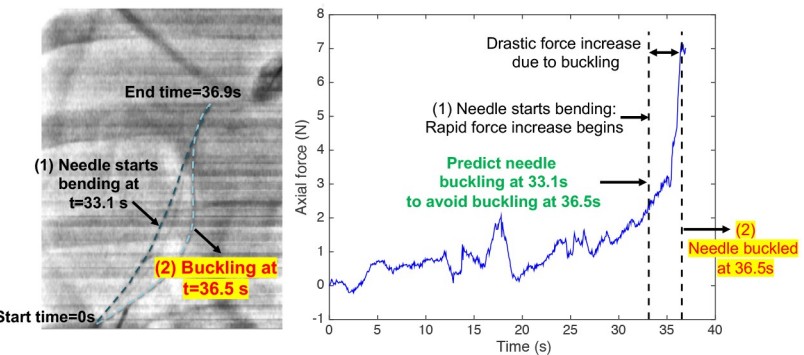

**Fig 1. Fluoroscopic motion image and force data plot of a needle buckling during insertions into a porcine cadaver liver [22].** The main idea is to predict buckling as soon as the needle starts bending slightly at 33.1s, before the needle completely buckles at 36.5s.

## 1.2 Related work

A brief survey of existing approaches to modeling and estimation of force data, prediction and control of adverse events related to robot-assisted needle steering domain, and predictive models using forecasting and adaptive control strategies, is presented below.

**1.2.1 Force modeling and estimation.**   Over the past decade, researchers have begun to realize the importance of estimating force data and providing control to a surgeon for timely identification and prevention of unforeseen incidents during surgical procedures with a robot [23, 24]. With this goal, several insertion force models were developed using analytical techniques [25, 26]. The high computation efficiency of the analytical techniques make them suitable candidates for online force estimation and control [27]. In most of these techniques, the force data were decomposed into friction, cutting and stiffness force components and each component was modeled independently. For example, Simone estimated the friction and stiffness forces using modified Karnopp and nonlinear spring models, respectively, with the models being capable of providing force feedback in real time [25]. With the nonlinear spring model, puncture of the interior tissue structures was controlled online while detecting undesired collisions. For cutting force modeling, a novel method was proposed using force data patterns to estimate the local elastic modulus and friction state of the needle-tissue interactions in real time [28]. As a measure to ensure patient safety, Fukushima *et al*. estimated the cutting force from the friction force obtained through force visualization so that human errors were reduced during needle insertion [29]. Another method which directly modeled the force data without decomposition was a multilayer model [26]. Their method was tested through different layers of the tissue (fat, skin, muscle) with the main objective of force control to achieve optimal puncture tasks. Since these analytical models are just approximations of the original force data, it is difficult to capture unexpected force profiles due to an adverse event.

There has been growing interest in data-driven methods for force estimation [15]. For example, one of the first works to use only insertion force data to identify cancerous tissues in real time was [30]. Using the variation in the force patterns, they identified a threshold force value which classified cancerous tissues from normal tissues. Next, Gerwen *et al*. developed a stochastic model using only measured force data, to identify relevant peak forces required to puncture through a kidney [31]. Recently, deep neural network based models have also been developed to estimate insertion force without prior knowledge of needle-tissue characteristics. Gessert *et al*. estimated needle-tip forces by mapping insertion forces from a fiber-optical

sensor and the optical coherence tomography (OCT) using deep convolutional neural networks [32]. Their method demonstrated real-time capability due to low computation times. Since high resolution imaging modalities such as OCT are expensive for the operating room, simple, inexpensive, and effective methods are preferred for accurate needle insertions [33].

The common goals of the above models were to estimate force data in real time for tissue characterization, detect events such as a puncture on occurrence, and accordingly optimize needle insertions. However, there has not yet been an attempt to estimate force data for future time horizons and predict puncturing events before they actually occurred. In general, there is tremendous need to optimize patient safety through real-time prediction and prevention of an adverse event before occurrence [8].

**1.2.2 Prediction and control of adverse events.** There are few examples in the literature which largely focus on controlling adverse events such as needle deflections and tissue deformations during insertions into a tissue, using only sensor data. For example, Rossa *et al.* estimated the needle-tip deflections using only sensor force data, needle-tip positions from 2D ultrasound, and steering inputs [34]. Using learning models from past measurements, optimal steering inputs were determined to avoid future deflections. Their models showed real-time capability with low computation time of 13ms. Next, Buzurovic *et al.* minimized tissue displacements and optimized needle insertions using a combination of artificial neural networks (ANN), neural-network based predictive control (NNPC) and model predictive control (MPC) techniques [35]. Their approach involved; a) estimating insertion force patterns with ANN, b) predicting and minimizing reactive forces using NNPC, and c) determining insertion inputs using NNPC and MPC to minimize tissue displacements. However, the minimum prediction computation time obtained was 300ms which might be challenging for real-time implementation.

Several strategies have been implemented in the past to avoid needle buckling events within the tissue. For example, novel designs for steerable needles were developed to increase the critical buckling force of the needle shaft and eventually reduce insertion force on the needle during insertions [36–38]. These methods only decrease the possibility of needle buckling events; however, they do not provide for real-time control mechanism when a needle buckling event does occur. To identify a needle buckling within the tissue, Tang *et al.* derived a relationship between the insertion forces and needle displacements using needle-tissue biomechanics model [39]. Other examples of modeling and detecting needle buckling events with biomechanics models include [40, 41]. However, with these methods, prior information of tissue properties was required, which is challenging for real-time implementation when tissue environments are generally unknown.

Recently, Li *et al.* developed a model-free adaptive control based on the Kalman filter technique to track and control the position of a flexible continuum robot [42]. Their approach did not use any prior information of the robot kinematics or tissue properties, and guaranteed real-time capability with computation time as low as 2ms. Additionally, they provided an optimal control law to predict a potential buckling of the continuum robot. This was the only closest work which used model-free adaptive techniques to control and predict a buckling based on some optimal control law. However, the main focus of their work was to track needle-tip positions in real time. To the best of our knowledge, there has not yet been an estimate or forecast in real time, of future values of insertion forces to predict needle buckling before it occurs.

**1.2.3 Forecasting and adaptive control.** In many prediction models, a combination of forecasting and adaptive control techniques is used such that the controller computes optimal decisions for prediction based on the forecasted data [43–46]. For example, a model predictive controller was used to compute the energy costs spent to keep the room temperature at comfortable levels using weather forecasts [43]. Wang *et al.* used two forecasting algorithms such as

Holt-Winters and Gaussian process to estimate the expected and stochastic demands of drinking water [46]. Then, predictive control of drinking water networks is performed using the forecasted demands as disturbance inputs. Recently, [47] proposed a gaussian process based adaptive Gaussian mixture model (GP-aGMM) to predict driver behavior and vehicle motion at road intersections and anticipate the motion of a dynamic obstacle with an intelligent controller. Their model consists of a gaussian process regression technique for probabilistic prediction of the driver behavior, GP-aGMM for accurate and faster prediction, and integration with an intelligent controller for closed-loop performance and safety. Their method used only raw sensor data (i.e., driver and vehicle positions), and did not require prior modeling of the driver and controller dynamics. The common feature of the above models was that the forecasting/ prediction and control strategies were implemented as complementary techniques. However, developing a forecasting model directly from a control strategy is yet to be explored.

## 1.3 Contributions

There are various existing techniques in the statistical and machine learning literature to perform forecasts for time series data; however their limitations of high computational complexity make real-time predictions of critical events difficult. To address this limitation, we derive a novel Compact Form Dynamic Linearization Model-Free Prediction (CFDL-MFP) method for time series forecasting, based on the existing dynamic linearization technique originally meant for feedback control problems [48]. The contribution of our current work is thus two fold: 1) develop CFDL-MFP to forecast time series data, and 2) develop a critical event prediction algorithm using forecasted data.

The paper is structured as follows: Section 2 discusses limitations of the existing time series forecasting methods with respect to real-time predictions of insertion force data. Section 3 proposes the novel time series forecasting model including analysis of model stability and forecast efficiency. Section 4 discusses simulation results of the proposed forecast as precursor to buckling prediction. Section 5 develops a needle buckling prediction algorithm using the proposed forecasts. Section 6 presents experimental results of the proposed algorithms in artificial and biological tissue. Section 7 concludes the paper.

## 2 Time series forecast: State of the art

Forecasting is a technique of projecting future values of a time series data depending on a prediction horizon. One-step forecast is performed if the immediate future value is estimated (prediction horizon = 1) and a multi-step forecast is performed if more than one future values are estimated. A brief review of some useful state-of-the-art statistical and machine learning (ML) techniques is presented in terms of their limitations with respect to real-time prediction of insertion force data. The main focus is to motivate the development of the proposed CFDL-MFP forecast, and choose a benchmark algorithm for performance analysis.

## 2.1 Statistical models

Classical or statistical time series forecasting are the oldest and most sophisticated linear based techniques that work well on a wide range of data. Useful models such as Random Walk, Theta, Exponential Smoothing (ETS), and Autoregressive Integrated Moving Average (ARIMA) algorithms are discussed.

**2.1.1 Random walk.** The most basic forecasting technique adds a random step to the last observed value. This characteristic of randomness makes the model a baseline for comparing the performance of other forecasting methods. However, it is nearly impossible to predict future values based on past trends due to uncertainty [49]. Since insertion force data generally

consist of global trends due to needle-tissue friction, random walk model is not useful for both one-step and multi-step force predictions.

**2.1.2 ETS.** The method predicts future values of a time series data using the weighted sum of the historical data in which the weights exponentially decrease as the past data become older. The model is simple and works well on all data since the parameters are obtained from numerical optimization. The key advantage is fast computation time as only few observed data points are required for prediction. The main limitation is that forecast accuracy gets worse for multi-step prediction [50], which is not suitable for our application.

**2.1.3 Theta.** Forecasting is performed by decomposing past data into linear combinations of second differenced approximations with appropriate choice of a parameter called 'theta' [51]. The idea was to capture both long and short term dependencies for forecast. For example, lower values of theta (zero theta models a linear trend) capture long term dependency, while higher theta captures short term dependencies. Forecast is performed using these two or more theta lines by choosing appropriate values of theta. This method was shown to have the best multi-step forecast accuracy [52]. A major drawback is that lot of hand tuning is required to select theta lines which increases complexity as size of the data increases. Later, an optimized method of choosing theta was proposed to use only relevant data for forecast [53]. Their optimized model promises high forecast accuracy compared to the original theta method. However, every profile of force data is important as it characterizes needle-tissue interactions. Thus, optimal theta model cannot be used to predict force data.

**2.1.4 ARIMA.** The method uses a linear combination of the previous data forming an autoregressive (AR) component, and a linear combination of the previous forecast errors forming a moving average (MA) component to perform time series forecasting [54]. Autocorrelation patterns of the previous data are used to identify the orders of AR and MA. Most importantly, stationarity in the data should be ensured for the method to work well. Thus, the basic forecasting steps include:

1. Identify the number of differencing operations ($d$) required to transform original time series into stationary data,

2. Identify the order (number) of AR components ($p$) and MA components ($q$) from the autocorrelations,

3. Obtain optimal coefficients of AR and MA through model-fitting, and

4. Obtain forecasts using these optimal coefficients for a given prediction horizon.

High forecast accuracy is achieved if the parameters are properly tuned, which requires a good understanding of the data. Due to robustness against noise, the model is most widely used for analyzing and forecasting many time series data. However, the computation time for forecast is medium to high depending on the nature of the data. For example, computation time is low if the data is already stationary. The computation time is high if the data are nonstationary due to the differencing operations required to achieve stationarity. With this limitation, ARIMA cannot be used for real-time force prediction.

## 2.2 ML models

Machine learning techniques use non linear models to analyze and forecast time series data. In general, supervised ML is implemented for forecasting where the models iteratively learn from previous observations to obtain good approximations and accurate forecasts of the data. Useful forecasting techniques include parametric (neural network based) models such as Multi-Layer Perceptron (MLP), Long Short-Term Memory (LSTM), and non-parametric models such as

Deep Gaussian Process (DGP) and K-Nearest Neighbor (KNN). Parametric methods perform accurate forecasts since back-propagation is involved in the training phase to obtain optimal model parameters, requiring very high computation power. For example, MLP has the best forecast accuracy at the cost of high computation requirement [55]. Time series data are intrinsically non-stationary with dynamic statistical distribution due to which retraining of the model is necessary for every entry of a new observation requiring high computational power [56]. Secondly, LSTM was shown to have the worst forecast accuracy [52]. These limitations currently make parametric ML unsuitable for real-time forecasting applications.

Non-parametric methods eliminate the above back propagation steps since they do not use prior statistical distribution of the data for forecasting. This results in less computation complexity at the cost of low accuracy. However, the requirement for training data increases as data grows in size resulting in high computation time [57]. Recent research has shown to improve forecast accuracy and optimize computation power through KNN and DGP models. KNN had less computation complexity with computation time of 0.41 second per training iteration, and the forecast accuracy was comparable (6% lower) to MLP [58]. Secondly, DGP had the least computation time of 0.36 second for each training iteration [59]. The overall forecast computation times of KNN and DGP would be much higher depending on the number of training iterations, making even non-parametric ML challenging for real-time prediction of force data. Since computation times are desired to be less than data sampling intervals (typically in tens of milliseconds), existing ML methods cannot be used for real-time force prediction.

A summary of the strengths and limitations of statistical and ML methods is given in Table 1. ML has highest computation times and lower forecast accuracy then statistical methods [52]. Thus, ML is not suitable for our application. Within the statistical methods, ARIMA is the most robust forecasting technique which finds vast applications in medical healthcare systems [60]. Hence, ARIMA is chosen as a benchmark to evaluate our CFDL-MFP method introduced in the next section.

## 3 Force prediction model

The force forecast model CFDL-MFP is derived by modifying the model-free control technique first introduced by [48]. Their control methodology is data-driven or model-free in the sense that only system input-output information are used to design feedback controllers for a general class of nonlinear systems with unknown dynamics. Their model entails predicting a future system output, $y(k + 1)$ using; a) system output at current time, $y(k)$, b) some future reference (desired) signal, $y_{ref}(k + 1)$, and c) control input, $u(k)$ obtained as a function of $y(k)$ and $y_{ref}(k + 1)$ so that the system produces desired outputs with $y(k + 1) \approx y_{ref}(k + 1)$. Similar to the standard predictive feedback control, $y_{ref}(k + 1)$ is defined beforehand (either manual or

**Table 1. Classical vs ML models.**

| Models | Strengths | Limitations | References |
|---|---|---|---|
| Random walk | Simplest and fastest computation | Not suitable for global trends | [49] |
| ETS | Fast computation time | Worse multi-step forecast accuracy | [50] |
| Theta | Best forecast accuracy | Heavy parameter tuning, complex computation | [53] |
| **ARIMA** * | High accuracy depending on $(p, d, q)$, Robust | Medium to high computation time depending on $d$ | [54] |
| ML | Best data fitting due to nonlinear modeling | Poor forecast accuracy, | [52] |
| | | Highest computation requirements | [57] |

* baseline model for comparison with proposed CFDL-MFP

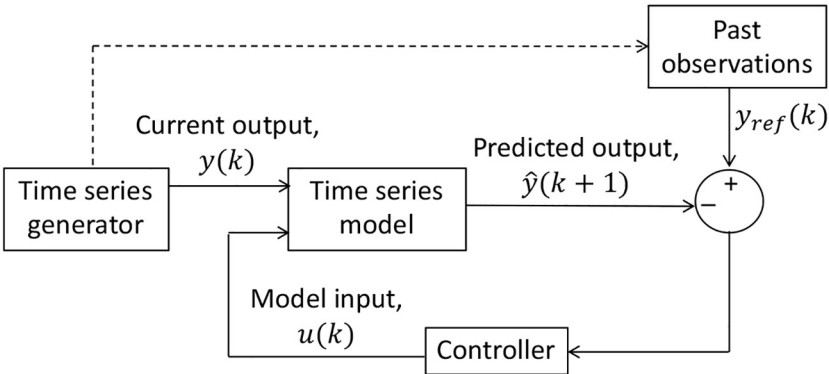

**Fig 2. CFDL-MFP model for time series forecast.**

system dynamics) to ensure convergence of $y(k + 1)$. For forecasting applications however, $y_{ref}(k + 1)$ becomes impossible since the underlying behavior of a time series data is observed using only past and current observations. Moreover, forecasting is already achieved if $y_{ref}(k + 1)$ is identified from the past observations. Replacing $y_{ref}(k + 1)$ with $y_{ref}(k)$ as a function of the previous observations, the novel time series forecast CFDL-MFP is developed as shown in Fig 2. Note that for time series models, the predicted output is denoted as $\hat{y}$ to differentiate from the actual observations, $y$.

Since the time series model is the one-dimensional axial force data generated from a force sensor, $\vec{f}_a$, the derivation of CFDL-MFP will be specific to the force data. Referring to Fig 2, the components of CFDL-MFP for force prediction include:

1. Force data as a time series model,

2. Force measured from a sensor as the current system output, $y(k) \leftarrow \vec{f}_a(k)$,

3. Reference force, $y_{ref}(k) \leftarrow$ past observations of $\vec{f}_a$,

4. Model input, $u(k)$ obtained from $y_{ref}(k)$ and previous prediction, $\hat{y}(k)$, and finally

5. Predicted forces, $\hat{y}(k + 1)$ obtained from $y(k)$ and $u(k)$ using the time series model.

For simplicity, the terms *forecast* and *prediction* will be used interchangeably in this paper.

### 3.1 CFDL-MFP

Let $\vec{f}_a(k) \in \mathbb{R}$ be the axial force measured from a force sensor at time $k$. Due to inherent measurement noise and lack of prior knowledge of tissue characteristics, the force data is modeled as a nonlinear time series function of the form:

$$\vec{f}_a(k + 1) = \mathbf{F}(\vec{f}_a(k), u(k))  \tag{1}$$

where $\mathbf{F}$ is some unknown nonlinear function. In order to obtain a good estimate of $\mathbf{F}$, we need to have a good understanding of the relationship between the axial forces, tissue properties, and user-defined inputs (for eg. needle insertion velocity and steering direction). Since tissue environments are unknown, it is complicated to establish a direct relationship between the forces and the inputs in real time. Recently, [61] used prior training data to learn the mapping

between the forces and user inputs. However, performing multiple insertions to learn the input-output mapping is time consuming. Due to these challenges, accurate modeling of the nonlinear function is difficult. By compressing the nonlinearities of Eq (1) into a single time-varying scalar known as the pseudo partial derivative (PPD), $\phi_c(k)$, a virtual dynamic linear model is realized to predict force data as shown:

$$\vec{f}_a(k+1) = \vec{f}_a(k) + \phi_c(k)\Delta u(k) \tag{2}$$

The model resembles a discrete-time SISO system with $\Delta u(k) = u(k) - u(k-1)$ as the incremental system input, and $\Delta \vec{f}_a(k+1) = \vec{f}_a(k+1) - \vec{f}_a(k)$ as the incremental system output. PPD is obtained by taking the partial derivative of the nonlinear function **F** with respect to $u(k)$ [48]. Since all the compressed nonlinearities are contained in $\phi_c(k)$, it is challenging to obtain an accurate value of $\phi_c(k)$ through analytical techniques. Hence, PPD is estimated numerically as $\hat{\phi}_c(k)$. Re-writing Eq (2) in terms of the estimated values with notation (), the force prediction model now becomes:

$$\hat{f}_a(k+1) = \hat{f}_a(k) + \hat{\phi}_c(k)\Delta u(k) \tag{3}$$

The model performs a 1-step ahead force prediction since the terms at each time state $k$ are scalar. If each term in Eq (3) contains a vector of values from the previous 2 states $(k-1, k)$, then 2-step prediction is performed. Similarly, if each term contains a vector of values from previous $N$ states $(k-N+1, .., k)$, then $N$-step ahead prediction is performed, where $N \geq 1$ is an integer known as the prediction horizon. The general $N$-step ahead force prediction model is:

$$\hat{F}_N(k+1) = \hat{F}_N(k) + \hat{\Phi}_N(k) \circ \Delta U(k) \tag{4}$$

where,

$$\hat{F}_N(k+1) = \begin{bmatrix} \hat{f}_a(k+1) \\ \hat{f}_a(k+2) \\ \vdots \\ \hat{f}_a(k+N) \end{bmatrix}_{Nx1}, \hat{F}_N(k) = \begin{bmatrix} \hat{f}_a(k-N+1) \\ \vdots \\ \hat{f}_a(k-1) \\ \hat{f}_a(k) \end{bmatrix}_{Nx1}$$

$$\hat{\Phi}_N(k) = \begin{bmatrix} \hat{\phi}_c(k-N+1) \\ \vdots \\ \hat{\phi}_c(k-1) \\ \hat{\phi}_c(k) \end{bmatrix}, \Delta U_N(k) = \begin{bmatrix} \Delta u(k-N+1) \\ \vdots \\ \Delta u(k-1) \\ \Delta u(k) \end{bmatrix}$$

and $\circ$ refers to the matrix element-wise operation. Note that $\hat{F}_N(k+1)$ can be obtained only when $\Delta U_N(k)$ and $\hat{\Phi}_N(k)$ are computed. In order to derive $\Delta U_N(k)$, we first define the following two reference force vectors:

$$\vec{F}_N^*(k) = \left[ \vec{f}_p^*(k-N+1), \ldots, \vec{f}_p^*(k) \right]^T$$

$$\vec{F}_{aN}(k) = \left[ \vec{f}_a(k-N+1), \ldots, \vec{f}_a(k) \right]^T \tag{5}$$

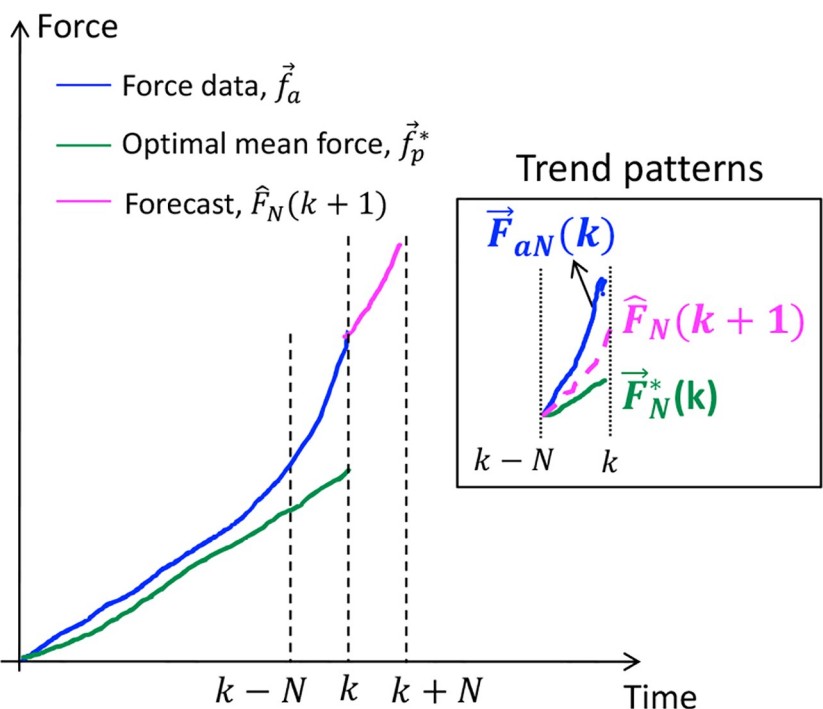

**Fig 3. Forecasts, $\hat{F}_N(k+1)$ obtained from optimal utilization of reference forces, $\vec{F}_N^*(k)$ and $\vec{F}_{aN}(k)$.** The rectangular box shows the trend patterns of the forecast bounded between the trends of the reference forces.

The first reference force, $\vec{F}_N^*(k)$ is a vector of previous $N$ values of the optimal mean forces, $\vec{f}_p^*$ generated from the past $(k-1)$ values of the force data. More details about the procedure to obtain $\vec{f}_p^*$ are found in our previous work [21]. Thus, $\vec{F}_N^*(k)$ represents the general trend pattern of the previously observed force data. The second reference force, $\vec{F}_{aN}$ represents the most recent force data. Since our force prediction model depends on the past and current force measurements, the trend pattern of a forecast is assumed to be bounded between the trends of $\vec{F}_N^*(k)$ and $\vec{F}_{aN}(k)$, shown in Fig 3. This assumption ensures that the forecast is always bounded and does not increase or exaggerate beyond the previously observed force trends. If the trend patterns of $\vec{F}_{aN}(k)$ and $\vec{F}_N^*(k)$ are different, the trend of the forecast will converge to some optimal trend between the reference trends. However, if the reference force trends are similar, then the trend of the forecasts will converge to the reference trends. For this purpose, optimal model input $\Delta U_N(k)$ is derived from a standard quadratic cost function, $J$ expressed as a function of the errors between the predicted output, $\hat{F}_N(k+1)$ and the reference forces:

$$J(\Delta U_N(k)) = |\vec{F}_N^*(k) - \hat{F}_N(k+1)|^2 \quad +|\vec{F}_{aN}(k) - \hat{F}_N(k+1)|^2$$
$$+\lambda|\Delta U(k)|^2 \tag{6}$$

Substituting $\hat{F}_N(k+1)$ from (4) in (5), we have,

$$J(\Delta U_N(k)) = |\vec{F}_N^*(k) - \hat{F}_N(k) - \hat{\Phi}(k) \circ \Delta U_N(k)|^2$$
$$+|\vec{F}_{aN}(k) - \hat{F}_N(k) - \hat{\Phi}(k) \circ \Delta U_N(k)|^2$$
$$+\lambda|\Delta U(k)|^2$$

Optimal $\Delta U_N(k)$ is obtained by taking the partial derivative of $J$ with respect to $\Delta U_N(k)$ and setting to zero:

$$\Delta U_N(k) = \frac{\hat{\Phi}_N(k)}{(\lambda.\vec{\mathbf{1}} + 2\hat{\Phi}_N^2(k))} \circ \left(\vec{F}_N^*(k) + \vec{F}_{aN}(k) - 2\hat{F}_N(k)\right) \tag{7}$$

where $\vec{\mathbf{1}}$ is a $N$ dimensional vector of 1's and $\Delta U_N(k) \in \mathbb{R}^{N \times 1}$. The weighting factor $\lambda > 0$ restricts drastic changes in $\Delta U_N(k)$. Large values of $\lambda$ cause $\Delta U_N(k) \to 0$ resulting in poor forecasts $|\hat{F}_N(k+1) - \hat{F}_N(k)| \to 0$ from Eq (4), showing that the reference forces are not utilized efficiently for prediction. Small $\lambda$ cause $\Delta U_N(k) > 0$ which results in efficient forecasts through optimal utilization of the reference forces. Therefore, $\lambda$ is an important tuning parameter for accurate force prediction.

The prediction model is robust if it is not affected by dynamic changes in PPD, as follows:

$$\begin{aligned}
\text{If } \hat{F}_N(k) &= \hat{F}_N(k-1) + \hat{\Phi}_N(k-1) \circ \Delta U_N(k-1) \\
\text{and } \tilde{F}_N(k) &= \hat{F}_N(k-1) + \hat{\Phi}_N(k) \circ \Delta U_N(k-1) \\
\text{then } \hat{F}_N(k) &\simeq \tilde{F}_N(k) \text{ only if } \hat{\Phi}_N(k-1) \simeq \hat{\Phi}_N(k)
\end{aligned} \tag{8}$$

The error in the forecasts, $|\hat{F}_N(k) - \tilde{F}_N(k)|$ due to changes in PPD should be minimized for robustness. Next, $|\hat{\Phi}_N(k) - \hat{\Phi}_N(k-1)|$ should be minimized to transform PPD into a slowly time-varying scalar. For this purpose, the quadratic cost function $J$ is expressed as a function of these errors:

$$\begin{aligned}
J(\hat{\Phi}_N(k)) = &|\hat{F}_N(k) - \hat{F}_N(k-1) - \hat{\Phi}_N(k) \circ \Delta U(k-1)|^2 \\
&+ \mu|\hat{\Phi}_N(k) - \hat{\Phi}_N(k-1)|^2
\end{aligned} \tag{9}$$

PPD is estimated by taking the partial derivative of $J(\hat{\Phi}_N(k))$ with respect to $\hat{\Phi}_N(k)$ and setting to zero:

$$\begin{aligned}
\hat{\Phi}_N(k) = &(\mu.\vec{\mathbf{1}} + \Delta U_N(k-1))^{\circ-1} \circ (\mu\hat{\Phi}_N(k-1) \\
&+ \Delta U_N(k-1) \circ \Delta\hat{F}_N(k))
\end{aligned} \tag{10}$$

The weighting factor $\mu > 0$ applies a penalty on the time-varying change in $\hat{\Phi}_N$. For example, PPD is constant for large $\mu$, and slowly time-varying for small $\mu$. Finally, $N$-step ahead prediction of force data is performed iteratively using Eqs (4), (7) and (10).

## 3.2 Model stability and forecast analysis

Stability of Eq (4) depends on the boundedness of the forecast trends with respect to the reference force trends. For example, if the forecast trend exceeds the reference force trends, exaggerated forecasts occur as shown in Fig 4, and prediction model is considered unstable. This is undesirable from the energy conservation law since any change in the system output, $|\hat{F}_N(k+1) - \hat{F}_N(k)|$ cannot exceed a finite change in the system input, $|\Delta U_N(k)|$, which leads us to state the first condition of stability.

**Stability Condition 1**. *Stability is guarenteed if for every bounded reference force, $(\vec{F}_N^*(k), \vec{F}_{aN}(k))$, and bounded model input with $U_N(k) \neq U_N(k-1)$, there exists a bounded model*

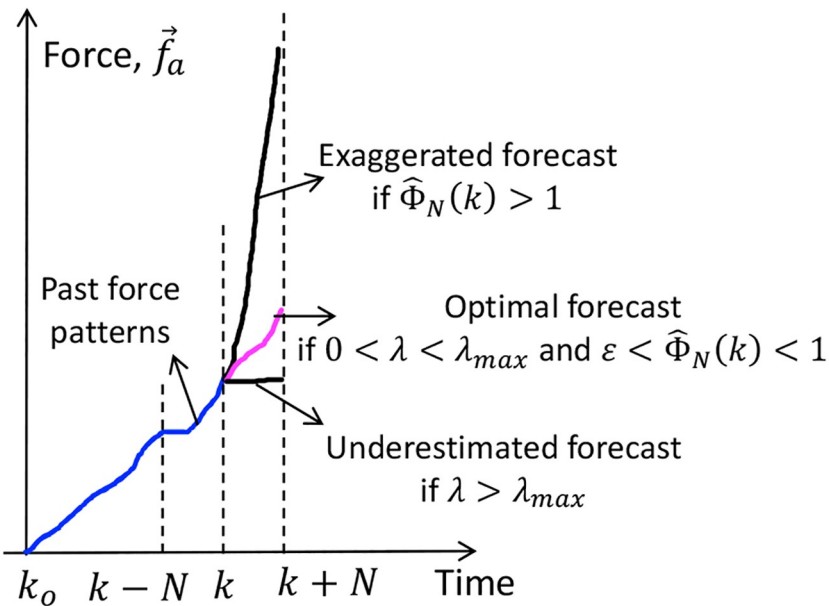

**Fig 4. Ideal plot showing exaggerated, underestimated and optimal forecasts with varying $\hat{\Phi}_N(k)$ and λ.**

*output, $\hat{F}_N(k+1)$ such that*

$$|\hat{F}_N(k+1) - \hat{F}_N(k)| \leq |U_N(k) - U_N(k-1)| \tag{11}$$

*for every $k \geq 0$. This condition is also called Bounded Input Bounded Output (BIBO) stability.*

To ensure stability, it is necessary to establish the boundedness condition of PPD. The next condition states the limiting value of PPD for stability.

**Stability Condition 2**. *There exists a maximum $\hat{\Phi}_N(k) = \Phi_{max}$ above which prediction model becomes unstable. That is,*

$$If \quad \hat{\Phi}_N(k) \quad > \Phi_{max}, \ then$$
$$|\hat{F}_N(k+1) - \hat{F}_N(k)| \quad > |U_N(k) - U_N(k-1)| \tag{12}$$

*and the limiting value is $\Phi_{max} = 1$.*

*Proof.* The proof is straightforward. From (4),

$$|\hat{F}_N(k+1) - \hat{F}_N(k)| \quad = |\hat{\Phi}_N(k) \circ \Delta U(k)|$$

Using Cauchy-Schwarz inequality,

$$|\hat{F}_N(k+1) - \hat{F}_N(k)| \quad \leq |\hat{\Phi}_N(k)| \circ |\Delta U_N(k)| \tag{13}$$

Comparing (10) and (12),

$$
\begin{aligned}
|\hat{F}_N(k+1) - \hat{F}_N(k)| \quad &< |\Delta U(k)| \text{ for } \hat{\Phi}_N(k) < 1 \;, \\
|\hat{F}_N(k+1) - \hat{F}_N(k)| \quad &= |\Delta U(k)| \text{ for } \hat{\Phi}_N(k) = 1 \text{ and} \\
|\hat{F}_N(k+1) - \hat{F}_N(k)| \quad &> \Delta U(k) \text{ for } \hat{\Phi}_N(k) > 1
\end{aligned}
\tag{14}
$$

Exaggerated forecasts can be avoided if the above stability conditions are satisfied. However, underestimated (poor) forecasts still occur for a stable model depending on the values of $\Phi_N(k)$ and $\lambda$ from Eq (7). The worst case of an underestimated forecast is $\hat{F}_N(k+1) = \hat{F}_N(k)$, as shown in Fig 4. The next two forecast conditions state the range of PPD and $\lambda$ required to prevent underestimated forecasts. The first condition states PPD for a fixed $\lambda$.

**Forecast Condition 1**. *For a given $\lambda > 0$, there exists a small constant $\varepsilon > 0$ such that* $|\hat{F}_N(k+1) - \hat{F}_N(k)| \to 0$ *if* $\hat{\Phi}_N(k) < \varepsilon$, *where $\varepsilon$ is very close to zero.*

*Proof.* For any $|U_N(k) - U_N(k-1)| \neq 0$, if $\hat{\Phi}_N(k) \to 0$ from Eq (4), then $|\hat{F}_N(k+1) - \hat{F}_N(k)| \to 0$.

Large values of $\lambda$ restrict the model input to be $\Delta U_N(k) \to 0$ which result in poor forecasts. For accuracy, it is necessary that $\Delta U_N(k) > \tilde{\varepsilon}$ is always satisfied, with $\tilde{\varepsilon}$ as a small positive constant close to zero. After satisfying *Forecast Condition 1*, the following condition states the maximum $\lambda$ for a fixed PPD.

**Forecast Condition 2**. *For a given $\hat{\Phi}_N(k) \in (\varepsilon, 1)$, there exists a maximum value $\lambda_{max}$ such that*

$$
\begin{aligned}
&If \;\; \lambda > \lambda_{max} \;\; then \;\; \Delta U_N(k) < \tilde{\varepsilon} \\
&and \;\; |\hat{F}_N(k+1) - \hat{F}_N(k)| \to 0
\end{aligned}
\tag{15}
$$

Finally, $\hat{\Phi}_N(k) < 1$ is required to achieve model stability, and $\hat{\Phi}_N(k) > \varepsilon$ and $\tilde{\varepsilon} < \lambda < \lambda_{max}$ are required for optimal forecasts. For our experiments, both $\varepsilon$ and $\tilde{\varepsilon}$ were chosen as $10^{-4}$.

## 3.3 Algorithm and architecture

The CFDL-MFP algorithm for iterative *N*-step force prediction is explained as a block diagram shown in Fig 5. The colored blocks represent the time state of each iteration with *k* as the current time, $k-1$ as the previous state, and $k+1$ as the future states within the prediction horizon. The starting time index is $k \geq 0$. Since previous *N* values of the force data are used for prediction, the first iteration begins from $k_o = N$, while ignoring the first *N* values. The pseudo code for iterative force prediction is presented in **Algorithm 1**. The inputs are force measurements, $\vec{f}_a(k)$, and a known prediction horizon, *N*. The output is the *N*-step ahead predicted force, $\hat{F}_N(k+1)$. Before iteration, some arbitrary values are assigned to the initial estimates, $(\hat{\Phi}_N(k_o), \Delta U_N(k_o), \hat{F}_N(k_o))$, and then $\hat{F}_N(k_o + 1)$ is predicted from these initial estimates using Eq (4), (steps 1-3). During each iteration, previous *N* values of the force data and optimal mean forces are assigned as reference forces, (steps 5-6). Next, the current estimates of PPD and model input are obtained as a function of their previous estimates, previous predicted forces, and the reference forces, (steps 7-8). The notations $E_4$, $E_7$, $E_{10}$ represent Eqs (4), (7) and (10), respectively. Finally, forecasts, $\hat{F}_N(k+1)$ are obtained from the current estimates following $E_4$, (step 9). The current values become previous estimates in the next iteration and this cycle continues till the end of the simulation time, $K_s$. Therefore, the algorithm can be implemented in real time.

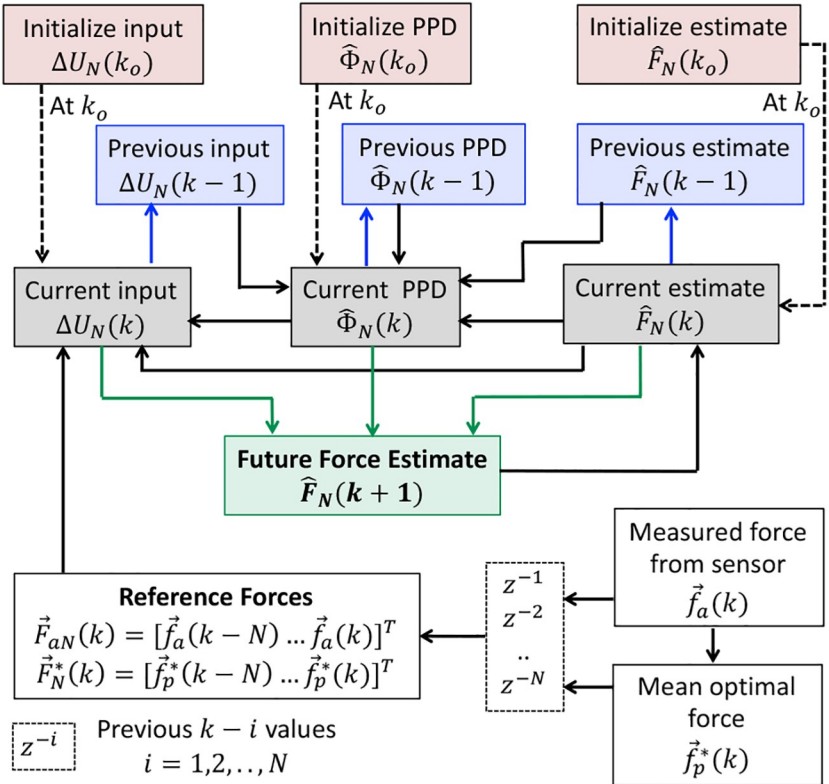

**Fig 5. Block diagram of the iterative CFDL-MFP algorithm for *N*-step ahead force prediction.** The block colors represent the time state for each iteration. Before iteration, the initial values of model input $\Delta U_N(k_o)$, PPD and output $\hat{F}_N(k_o)$ are arbitrarily assigned. In the first iteration as $k \leftarrow k_o + 1$, the initial values are used as current states (indicated by dotted arrows) to obtain the forecast for future states $k + 1$ (green arrows). In the next iteration with $k \leftarrow k + 1$, estimates from $k_o + 1$ become the previous estimates (blue arrows), and estimates from $k + 1$ become current estimates. Solid black arrows represent transition from previous to current states.

**Algorithm 1** CFDL-MFP

**Require:** Axial force, $\vec{f}_a(k)$, optimal mean force, $\vec{f}_p^*(k)$, and prediction horizon, $N$

1: **Initialize:** $k \leftarrow (k_o = N)$,
   $(\Delta U_N(k_o), \hat{\Phi}_N(k_o), \hat{F}_N(k_o)) \leftarrow$ arbitrary values
2: $\hat{F}_N(k_o + 1) \leftarrow E_4(\Delta U_N(k_o), \hat{\Phi}_N(k_o), \hat{F}_N(k_o))$
3: $k \leftarrow k_o + 1$
4: **while** $k < K_s$ **do**
5:   $\vec{F}_N^*(k) \leftarrow [\vec{f}_p^*(k - N + 1), .., \vec{f}_p^*(k)]$
6:   $\vec{F}_{aN}(k) \leftarrow [\vec{f}_a(k - N + 1), .., \vec{f}_a(k)]$
7:   $\hat{\Phi}_N(k) \leftarrow E_{10}(\hat{\Phi}_N(k - 1), (\Delta U_N(k), \hat{F}_N(k - 1), \hat{F}_N(k))$
8:   $\Delta U_N(k) \leftarrow E_7(\vec{F}_{aN}(k), \vec{F}_N^*(k), \hat{F}_N(k))$
9:   $\hat{F}_N(k + 1)) \leftarrow E_4(\Delta U_N(k), \hat{\Phi}_N(k), \hat{F}_N(k))$
10:   $k \leftarrow k + 1$
11: **end while**

In addition to the main simulation loop (steps 4-11), **Algorithm 1** does not require an explicit iteration loop for multi-step (*N*-step) force prediction since model Eq (4) is already in the *N*-dimensional vector form. This resembles the architecture of a direct multi-step forecast

strategy, with low computation time as the key feature [62]. Secondly, our proposed model utilizes both observed and previous predicted data for multi-step prediction with the observed data being the reference forces $(\vec{F}_{aN}(k), \vec{F}_N^*(k))$, and the predicted data being $\hat{F}_N(k)$. The technique of using observed and predicted data for forecast resembles the recursive predictive strategy with high forecast accuracy as the key feature [62]. The overall architecture of the CFDL-MFP resembles a hybrid direct-recursive forecast strategy which offers the benefits of the above strategies such as low computation time and high forecast accuracy [63].

## 4 Simulation methods and results

In this section, we present simulation methods and results of CFDL-MFP forecasts for force data corresponding to insertions in *ex vivo* human liver. The model is compared with benchmark ARIMA and evaluated for performance in terms of forecast accuracy, computation time and stability.

### 4.1 Methods

**4.1.1 Force data.**   To evaluate the CFDL-MFP algorithm, we used the needle insertion data from experiments with an *ex vivo* human liver [64]. The data consist of 1-D force vector, $\vec{f}_a$, and a vector of insertion time stamps, $\tau$ originally sampled for 1ms of time. We sampled the data at 5ms for faster computations while still retaining the original force profiles. The prediction horizon, $N$ is chosen depending on the data sampling interval, $\tau(k) - \tau(k-1)$,

$$N = \frac{t}{\tau(k) - \tau(k-1)} \tag{16}$$

where $t$ is the prediction time horizon in seconds. For example, if $\vec{f}_a$ is sampled for $\tau(k) - \tau(k-1) = 0.005$s and $t = 1.5$s, then $N = 300$ data points. Generally, larger time horizons lead to lower forecast accuracies for time series models [65], and therefore the effects of varying $t$ on the prediction performance will not be studied in this work. Typically, a short prediction horizon of $t = 1.5$s is chosen for accurate predictions [66].

**4.1.2 Performance metrics.**   The profiles of insertion force data vary due to tissue inhomogeneity, where each variation in the force characterizes needle-tissue interactions [67]. Thus, a good forecast accuracy should be ensured across all data points. For this purpose, a root mean square error (RMSE) metric that is sensitive to data fluctuations is chosen to evaluate forecast accuracy [54].

$$\text{RMSE} = \frac{1}{\sqrt{N}} ||\hat{F}_N(k+1) - \vec{F}_N(k+1)||_2 \tag{17}$$

where $||.||_2$ is the Norm-2 of the $N \times 1$ error vector between the predicted force, $\hat{F}_N(k+1)$ and the actual measured force, $\vec{F}_N(k+1) = [\vec{f}_a(k+1) \dots \vec{f}_a(k+N)]$. Since $\vec{F}_N(k+1)$ is not available during real-time implementation, RMSE is obtained when the entire force data, $\vec{f}_a$ are available at the end of the simulation.

The second performance metric is the computation time captured for each iteration while executing **Algorithm 1** through steps (5-10).

We define a third performance metric called the stability indicator, $\tilde{S}$ to check for BIBO stability of Eq (4), which can be computed as shown below:

$$\textbf{while } k < K_s \textbf{ do}$$

$$\text{If } |\Delta\hat{F}_i(k+1)| < |\Delta U_i(k)| \text{ for all } i = 1, 2, .., N$$

$$\text{then } S_i(k) \leftarrow \vec{\mathbf{1}} \in \mathbb{R}^i$$

$$\text{Else } S_i(k) \leftarrow \vec{\mathbf{0}} \in \mathbb{R}^i \tag{18}$$

$$\textbf{end while}$$

$$\tilde{S} = \frac{1}{K_s N} \sum_{k=0}^{K_s-1} \sum_{i=0}^{N-1} S_i(k) \; ; \; \tilde{S} \in [0, 1]$$

$$\text{If } \tilde{S} = 1 \text{ then BIBO stable}$$

where $S \in \mathbb{R}^{NK_s}$. The idea is to check if the incremental output $\Delta\hat{F}_i(k+1) = |\hat{F}_i(k+1) - \hat{F}_i(k)|$ is always less than the incremental input $\Delta U_i(k)$ for each prediction step $i$ within $N$ and iteration step $k$ until the end of simulation $K_s$. The stability flag, $S_i(k)$ is a vector of 1's with size $i$ if *Stability Condition 1* is satisfied at time $k$. The stability indicator, $\tilde{S}$ is equal to 1 if the total number of data points is equal to the summation of $S_i(k)$ across all $i$ and $k$. Therefore, $\tilde{S}$ is a scalar between 0 and 1, where 1 indicates model stability and 0 indicates instability.

The last metric, forecast indicator (FI) is a probabilistic measure of forecast efficiency. According to *Forecast Condition 1*, underestimated forecasts are a result of $|\hat{F}_N(k+1) - \hat{F}_N(k)| \to 0$, indicating low forecast efficiency. Thus, FI is calculated as the frequency of the outcomes $|\hat{F}_N(k+1) - \hat{F}_N(k)| \to 0$ occurred during the entire simulation:

$$\text{FI} = \frac{\Omega(|\hat{F}_N(k+1) - \hat{F}_N(k)| \to 0)}{K_s} \tag{19}$$

where $\Omega$ represents the number of outcomes, $K_s$ is total number of simulation steps. Again, FI$\in(0, 1)$ where FI$\to$1 causes low forecast efficiency.

**4.1.3 Parameter initialization.** The first $N$ values of a force sensor are ignored, and the initial time state is $k_o = N$. Zero initial values are assigned to $\Delta U_N(k_o)$ and $\hat{F}_N(k_o)$. The initial value of PPD, $\hat{\Phi}_N(k_o)$ is assigned any value between 0 and 1 according to *Stability Condition 2* and *Forecast Condition 1*. For convenience, these vectors are uniformly assigned as $\hat{\Phi}_N(k_o) = [a, .., a]^T \in \mathbb{R}^N$ where $0 < a < 1$. Going forward, $\hat{\Phi}_N(k_o) = a$ would implicitly refer to vector initialization. Similarly, the weighting parameter, $\lambda$ is assigned a constant between 0 and $\lambda_{max}$. By observation, $\lambda_{max}$ was found to be 100 for $\hat{\Phi}_N(k_o) = 0.5$. Therefore, $\hat{\Phi}_N(k_o) = 0.5$ and $\lambda = 0.1$ are chosen to first compare the performance of CFDL-MFP with auto-ARIMA. Secondly, the effects of varying PPD and $\lambda$ are studied. Next, $\mu = 1$ for convenience since PPD is already designed to be a slowly-time varying parameter according to Eq (10). Thus, any $\mu > 0$ does not affect forecast efficiency.

**4.1.4 auto-ARIMA.** The open source package *pyramid-arima 0.9.0* developed by the Python Software Foundation® was used as benchmark ARIMA to compare the performance of CFDL-MFP. Their *auto.arima* module automates the processing steps (1-3) mentioned in Section 2.1.4, and generates forecasts using previous data points and a known prediction horizon as inputs. ARIMA forecasting is performed iteratively and the related code snipper is

given in Algorithm 2. The functions **auto_arima** and **predict** are defined in the *auto.arima* package. Previous $N$ observations of the force data, $\vec{F}_N(k)$ are used to fit the ARIMA model, (steps 3-4). The output of the auto-ARIMA algorithm is the $N$-step forecast, $\hat{F}_N^{ar}(k+1)$ obtained from the **predict** function, (step 5).

**Algorithm 2** AUTO-ARIMA FOR FORCE DATA FORECAST

**Require:** Axial force, $\vec{f}_a(k)$, and prediction horizon, $N$
1: Initialize $k \leftarrow N$
2: **while** $k < K_s$ **do**
3: $\vec{F}_N(k) \leftarrow [\vec{f}_a(k-N+1),..,\vec{f}_a(k)]$
4: train $\leftarrow$ **auto_arima** $(\vec{F}_N(k))$
5: $\hat{F}_N^{ar}(k+1) \leftarrow$ **predict**(train, $N$)
6: $k \leftarrow k + 1$
7: **end while**

Replacing $\hat{F}_N(k+1)$ with $\hat{F}_N^{ar}(k+1)$ in Eq (17), forecast RMSE for auto-ARIMA is computed at the end of simulation. Similarly, computation time is captured for each iteration, while executing steps (3-6) of Algorithm 2.

The CFDL-MFP and auto-ARIMA algorithms were implemented in python 3.5.2 environment on a computer with the following features: Intel® Core i7-7700HQ CPU @2.8GHz, 16GB RAM, x64 based processor.

## 4.2 Forecast results and discussion

**4.2.1 CFDL-MFP forecasts.** The plot of forecasts for force data, $\vec{f}_a$ sampled at 5ms is shown in Fig 6(a). Even though forecasts are obtained at every iteration state $k$ from Algorithm 1, the forecasts at the following six key data points are shown to evaluate performance on different force profiles: (A = 4.86s, B = 14.24s, C = 18.65s, D = 52.85s, E = 74.6s, F = 76.15s), represented by dotted vertical lines. The plot of forecasts, $\hat{F}_N(k+1)$ and the observed forces, $\vec{F}_N(k+1)$ is zoomed at these six points for visibility. Each point represents a unique force profile, with a gradual increase at A and a flat force profile at D. There is a rapid increase of force around B, where B is chosen to lie between the start and end points of this rapid increase.

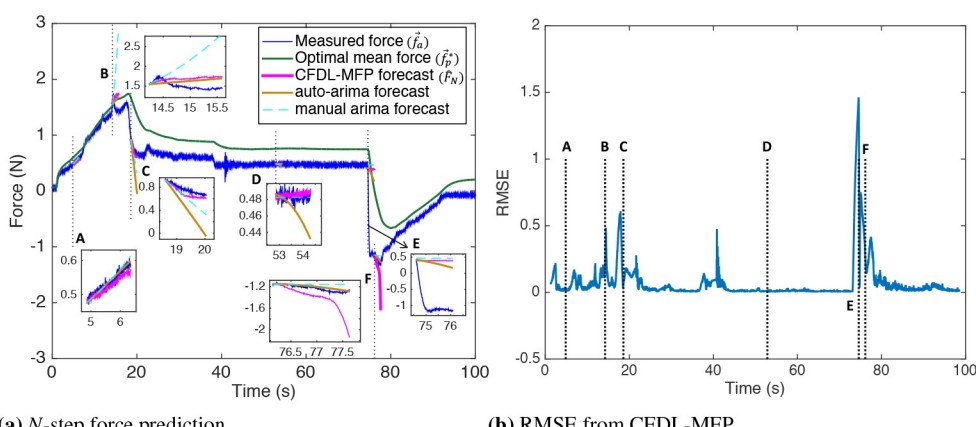

(a) $N$-step force prediction (b) RMSE from CFDL-MFP

**Fig 6. Forecasting results.** a) Plot of N-step force forecasts with CFDL-MFP (magenta), auto-ARIMA (dark orange), and manual ARIMA (dotted cyan) for force data sampled at 5ms. The CFDL-MFP parameters are initialized with $(\lambda = 0.1, \hat{\Phi}_N(k_o) = 0.5)$. b) Plot of RMSE showing spikes at rapidly changing force profiles. CFDL-MFP forecasts were most accurate for profiles (A,C,D).

Similarly, point C is chosen to lie between the start and end points of a rapid force decrease profile. Points (E,F) are chosen to lie at the intersection of rapidly changing profiles. The plot shows CFDL-MFP forecasts obtained for ($\lambda = 0.1$, $\Phi_N(k_o)$) = 0.5). The plot of RMSE obtained for each iteration is shown in Fig 6(b) with the dotted vertical lines representing the above six data points. Detailed analysis of the forecasts for each force profile is given below.

1. *Forecasts at k = (A,D)*: Since the force profiles were steady over long periods, the trend patterns of the reference forces were similar. As a result, the trend of the forecast converged to the reference trends. The forecasts were accurate with low RMSE since no rapid changes in the observed forces, $\vec{F}_N(k + 1)$ occurred, which is expected under normal needle steering conditions.

2. *Forecast at k = B*: The time interval between B and the starting point of rapid force increase was short to capture this rapid increase in $\vec{F}_N^*(k)$. For this reason, the trend of $\vec{F}_{aN}(k)$ was rapid, whereas $\vec{F}_N^*(k)$ was still gradual. Therefore, the trends of the forecast converged to an optimal trend bounded by the reference trends. RMSE was slightly high since the observed force rapidly decreased within the prediction horizon. This situation can lead to false predictions of a force increase, which will be discussed with respect to the prediction of needle buckling events in the later sections.

3. *Forecast at k = C*: The interval between C and the start point of the rapid force decrease was enough to capture this rapid decrease in $\vec{F}_N^*(k)$, resulting in similar reference trends. The trends of the forecast thus converged to both the reference trends. Since continuous decrease in force data is not realistic in needle insertions due to friction, rapid decrease in the observed forces, $\vec{F}_N(k + 1)$ stopped after point C. Since the reference forces consisted of both the rapid and general trends, the forecast was accurate with low RMSE.

4. *Forecasts at k = (E,F)*: Since forecasts are obtained from previously observed data, it is difficult to predict any unexpected force pattern without prior knowledge of the tissue properties. Therefore, the forecast RMSE was high at every intersection point of rapidly changing profiles (maximum at point E), shown in Fig 6(b).

**4.2.2 Comparison with auto-ARIMA.** The performance of CFDL-MFP was compared with auto-ARIMA in terms of RMSE and computation time for three sampling intervals: (5,20,800) ms, and the plot of auto-ARIMA forecasts, $\hat{F}_N^{ar}(k + 1)$ for 5ms sampling is shown in Fig 6(a) (dark orange). Exaggerated forecasts, $\vec{F}_N^{ar}(k + 1)$ occurred at C and D with the parameters automatically identified as ($p = 2$, $d = 1$, $q = 1$) and ($p = 1$, $d = 1$, $q = 1$), respectively. However, choosing $d = 0$ with trial and error produced better forecasts at D, shown as dotted cyan plot. Similarly, forecasts through manually tuned ARIMA parameters were plotted for all six profiles to show that auto-ARIMA did not always yield accurate forecasts. To obtain forecasts for all force profiles, manual tuning was required making real-time prediction challenging. Forecast RMSE values from auto-ARIMA and CFDL-MFP for six force profiles and three sampling intervals are given in Table 2. CFDL-MFP performed best forecasts at C and D, whereas forecast accuracies of both the algorithms were comparable for other force profiles and for all sampling intervals. Overall, the mean RMSE of CFDL-MFP was 11% lower than auto-ARIMA.

The computation times (mean, maximum, minimum) of the algorithms for each sampling interval are presented in Table 2. The maximum computation time for forecasting on 5ms sampled data was 49ms which is larger than the sampling interval, making real-time implementation challenging. However, such low sampling intervals are not practical since the CPU processing speed reduces considerably with too many data points. Very high sampling

**Table 2. auto-ARIMA vs CFDL-MFP.**

| | auto-ARIMA | | | CFDL-MFP | | |
|---|---|---|---|---|---|---|
| | sampling time (ms) | | | sampling time (ms) | | |
| | 5 | 20 | 800 | 5 | 20 | 800 |
| Data points | RMSE | | | | | |
| A | 0.009 | 0.036 | 0.037 | 0.010 | 0.052 | 0.085 |
| B | 0.172 | 0.180 | 0.209 | 0.218 | 0.225 | 0.412 |
| C | 0.385 | 0.529 | 0.776 | 0.071 | 0.061 | 0.209 |
| D | 0.024 | 0.061 | 0.007 | 0.006 | 0.008 | 0.004 |
| E | 1.313 | 1.209 | 1.636 | 1.405 | 1.225 | 1.631 |
| F | 0.035 | 0.286 | 0.065 | 0.248 | 0.245 | 0.060 |
| Minimum | 0.009 | 0.036 | 0.007 | 0.006 | 0.008 | 0.004 |
| *Mean (SD)* | *0.323 (0.505)* | *0.384 (0.442)* | *0.455 (0.646)* | *0.326 (0.538)* | *0.303 (0.462)* | *0.400 (0.620)* |
| Maximum | 1.313 | 1.209 | 1.636 | 1.405 | 1.225 | 1.631 |
| | Computation time (ms) | | | | | |
| Minimum | 4670 | 1970 | 156 | 0.10 | 0.015 | 0.017 |
| *Mean (SD)* | *27530 (25130)* | *4820 (2370)* | *603 (290)* | ***5.6 (3.4)*** | ***0.4 (0.3)*** | ***0.026 (0.078)*** |
| Maximum | 63720 | 7390 | 941 | 49 | 4.1 | 0.88 |

intervals such as 800ms are also not desired since valuable information in the data is lost. For sampling interval of 20ms, original force profile was retained and the maximum computation time was 4.1ms, which is ideal for real-time applications. On the other hand, auto-ARIMA showed very high computation times on the order of 4.8s on 20ms sampled data. As discussed earlier, the high computation time is due to the following reasons: a) data processing steps from Section 2.1.4 performed for each iteration, b) parameters identified for every data point to produce accurate forecasts, and c) a recursive strategy used for multi-step forecast [68]. The low computation time of CFDL-MFP is due to the following reasons: i) no data processing steps required as original data was used directly, ii) parameters $(\lambda, \hat{\Phi}_N(k_o))$ were initialized only once before iteration, and ii) a direct strategy used for multi-step forecast from Section 3.3. Compared to auto-ARIMA, CFDL-MFP was accurate, faster, and required less manual effort to perform multi-step forecasting.

**4.2.3 Stability analysis.** The stability plots for CFDL-MFP forecasts on 20ms sampled data are shown in Fig 7(a)–7(c). CFDL-MFP forecasts were performed independently for 4 different initial values of $\hat{\Phi}_N(k_o)$ to study their effects on stability. For each $\hat{\Phi}_N(k_o)$, the stability indicator, $\tilde{S}$ was computed according to Eq (18). For all $\hat{\Phi}_N(k_o) < 1$, $\tilde{S} = 1$ indicating model stability. For $\hat{\Phi}_N(k_o) = 1.01$, $\tilde{S} = 0$ indicating model instability, which in turn lead to exaggerated forecasts at every point as shown in Fig 7(b) (dotted red plot). Fig 7(c(i)) shows the plot of incremental model input, $|\Delta U_N(k)|$ and incremental forecasts, $|\hat{F}_N(k+1) - \hat{F}_N(k)|$ for $\hat{\Phi}_N(k_o) = 0.5$ where, BIBO stability was satisfied since $|\hat{F}_N(k+1) - \hat{F}_N(k)| < |\Delta U_N(k)|$ for all $k$. Next, PPD was constant at 0.5 throughout the simulation as shown in (c (ii)), owing to the slowly time-varying condition from Eq (10). Overall, any $\hat{\Phi}_N(k_o) < 1$ satisfy the Stability conditions defined in Section 3.2. Since any value of $\hat{\Phi}_N(k_o) < 1$ can be chosen to perform a stable prediction, the effect of $\Phi_N(k_o) = 1$ on forecasts is trivial and is not discussed.

**4.2.4 Forecast analysis.** With the weighting parameter, $\lambda$ fixed at 0.5, the forecast plots for $\Phi_N(k_o) = 0.001$ (solid dark orange), $\hat{\Phi}_N(k_o) = 0.01$ (dotted dark orange), and $\hat{\Phi}_N(k_o) = 0.5$ (solid dark green) are shown in Fig 7(b). The plot is zoomed in at profiles A and F containing

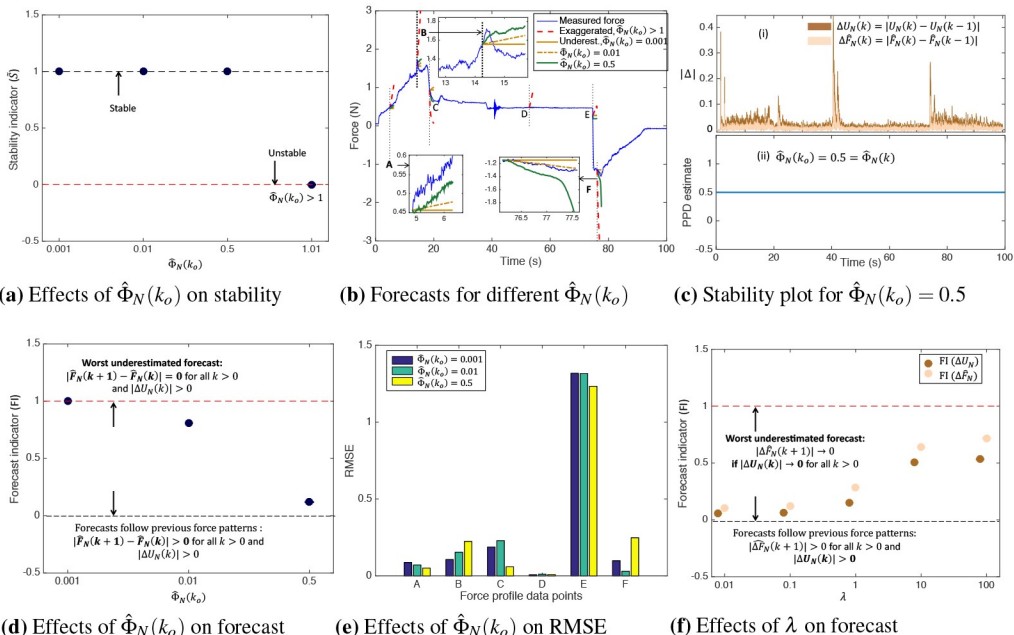

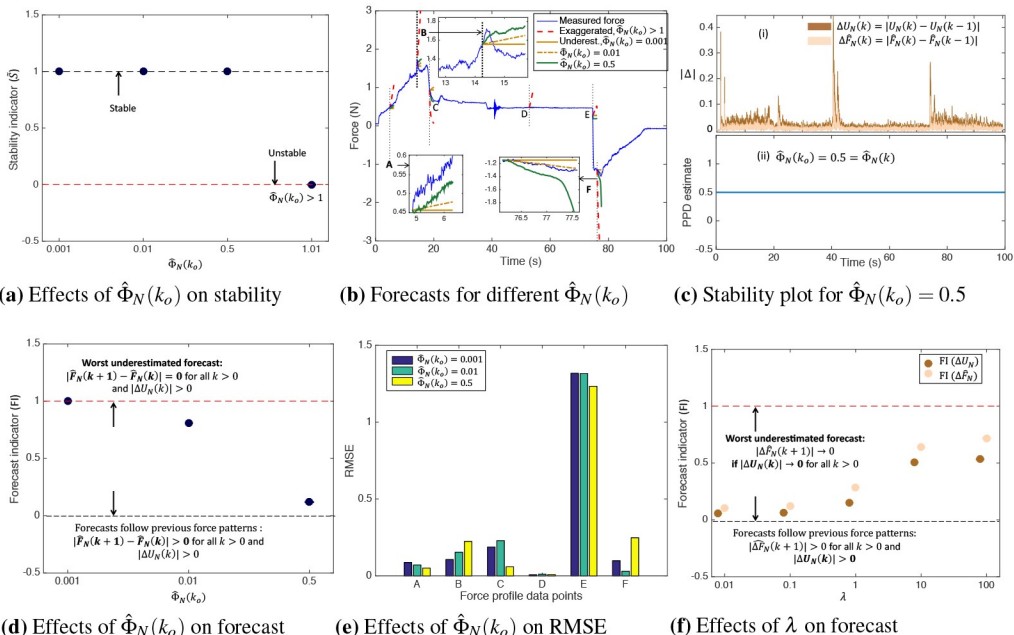

**(a)** Effects of $\hat{\Phi}_N(k_o)$ on stability **(b)** Forecasts for different $\hat{\Phi}_N(k_o)$ **(c)** Stability plot for $\hat{\Phi}_N(k_o) = 0.5$

**(d)** Effects of $\hat{\Phi}_N(k_o)$ on forecast **(e)** Effects of $\hat{\Phi}_N(k_o)$ on RMSE **(f)** Effects of $\lambda$ on forecast

**Fig 7. Plots showing stability and forecast analysis with varying $\hat{\Phi}_N(k_o)$ and $\lambda$.** a) Model is stable when $\bar{S} = 1$ for $\Phi_N(k_o) < 1$. b) Forecast plots for varying PPD with fixed $\lambda = 0.5$. PPD > 1 leads to exaggerated forecasts (dotted red plot), low PPD causes underestimated or linear forecasts (solid and dotted dark orange), and larger PPD provides nonlinear forecasts (dark green). c) (i) Input ($\Delta U_N$) and output ($\Delta \hat{F}_N$) plots satisfying *Stability Condition 1*, and (ii) constant PPD indicates validity of Eq (10). d) Increase in PPD decreases FI which makes forecasts more nonlinear. e) RMSE is low for linear forecasts at F (PPD = 0.01), and underestimated forecasts at B (PPD-0.001). However, RMSE was lowest for PPD = 0.5 at (A,D,E). f) FI increases with $\lambda$ causing underestimated forecasts. $\lambda \leq 1$ for nonlinear forecasts.

the forecasts and the observed force, $\vec{F}_N(k+1)$ over the prediction horizon. As visible from the zoomed plots, underestimated forecasts occurred for lower values of $\hat{\Phi}_N(k_o)$. According to the *Forecast Condition 1*, $\hat{\Phi}_N(k_o) < (\varepsilon = 0.001)$ was the minimum PPD at which underestimated forecasts occurred for the given $\lambda = 0.5$. As $\hat{\Phi}_N(k_o)$ increased to 0.01, the trends of the forecast slightly increased, but did not converge to the reference force trends and thus, appeared linear. This linearity was mainly due to lack of convergence with the reference trends, and is directly related to the forecast indicator (FI) which reduced when PPD increased as shown in Fig 7(d). As $\hat{\Phi}_N(k_o)$ increased to 0.5, FI $\rightarrow 0$, and the resulting forecasts converged to the reference trends. Sometimes, linear forecasts due to lack of convergence have higher accuracy as shown in Fig 7(e). The forecast RMSE at profile F was lowest for $\Phi_N(k_o) = 0.01$ since the forecasts resembled an accurate linear fit to the observed force data, evident in Fig 7 (b) (dotted dark orange). Otherwise, forecasts were accurate at other force profiles for $\hat{\Phi}_N(k_o) = 0.5$. Note that RMSE was lowest for $(\hat{\Phi}_N(k_o) = 0.001)$ at B, since the corresponding force profile resembled a Gaussian pattern with *zero mean* owing to rapid increase and decrease of similar magnitude, and the trends of the underestimated forecast resembled a constant fit to the observed force.

Similarly, the effects of varying $\lambda$ on the forecast efficiency are shown in Fig 7(f), keeping $\hat{\Phi}_N(k_o)$ fixed at 0.5. Underestimated forecasts occurred as $\lambda$ increased. From *Forecast Condition 2*, $\lambda_{max}$ was chosen as 10, since FI $\rightarrow 0$ and forecasts converged to the reference force trends when $\lambda < \lambda_{max}$.

**Table 3. Statistical analysis—Performance of CFDL-MFP.**

| Performance metric | Force profile | | Initial PPD, $\hat{\Phi}_N(k_o) < 1$ | | $\lambda > 0$ | |
|---|---|---|---|---|---|---|
| (Mean values) | $p$ | Significant Groups | $p$ | Significant Groups | $p$ | Significant Groups |
| RMSE | < 0.001 | (A,B,C,D,F)< E | 0.8912 | - | 0.99 | - |
| Computation time | < 0.001 | (A,B)<C<(D,E,F) | 0.3916 | - | 0.406 | - |
| Stability indicator, $\tilde{S}$ | - | stable if $\hat{\Phi}_N < 1$ | - | unstable for $\hat{\Phi}_N > 1$ | - | always stable |

## 4.3 Statistical analysis and discussion

The standard Kruskal-Wallis statistical analysis was performed (with significance level $p = 0.001$) by taking the average of the performance metrics across all force profiles and the CFDL-MFP parameters, $(\hat{\Phi}_N(k_o) < 1, \lambda > 0)$ for the sampling interval of 20ms, and the corresponding results are given in Table 3. Forecast RMSE was significantly high ($p < 0.001$) for profile E due to the unexpected rapid decrease. However, there was no significant effect of the parameters on the forecast accuracy. This showed that no value of $\hat{\Phi}_N(k_o) < 1$ or $\lambda > 0$ produced accurate forecasts at E. As length of data increased, the computation time significantly increased as expected. However, the maximum value of the computation time at the end of simulation was still 5 times lower than the sampling interval. Again, varying the parameters had no effect on the computation times. Finally, stability of the system was always ensured as long as $\hat{\Phi}_N(k_o) < 1$ irrespective of the force profiles and $\lambda$.

**4.3.1 Generalizability.** The results demonstrate the generalizability of CFDL-MFP since the selection of the parameters, $(\hat{\Phi}_N(k_o), \lambda)$ is a one-time process depending on the type of forecasts required. For example, underestimated or linear forecasts (with low PPD) were more accurate at intersection points of rapidly changing force profiles such as (E,F), from Fig 7. Similarly, forecasts converging to the previous force trends (with high PPD) were more accurate for stable force profiles (A,D) and rapidly changing profiles (B,C). If forecasts strictly following previous force trends are required, then $\lambda < \lambda_{max}$ and $\varepsilon < \hat{\Phi}_N(k_o) < 1$ can be chosen for any time series data. Since PPD does not change over time during simulations, exploring the correlation between $\lambda$ and PPD and the combined effects on forecasts is trivial. Tuning the parameters independently was enough to produce accurate forecasts. Therefore, $\Phi_N(k_o) = 0.5$ and $\lambda = 0.1$ are chosen to produce forecasts for axial force data and needle buckling prediction.

One main limitation of the proposed forecast is that the number of previous observations used as reference force data (i.e reference horizon) was always equal to the prediction horizon $N$, see Section 3.1. Restricting the reference horizon to $N$ provided reasonably accurate forecasts for many force profiles as shown from the above results. Sometimes, different reference horizons are still preferred to ensure accuracy depending on certain force profiles. For example, if periodic fluctuations in the data are common, then reference horizons greater than $N$ are preferred so the general patterns are captured to provide reasonable forecasts. For this purpose, a slight modification in the vector operations is required to ensure validity of the Eqs (7) and (10). Developing a constrained optimization model with adaptive tuning of CFDL-MFP parameters to provide accurate forecasts at all times, and exploring the effects of reference horizon on forecast accuracy would be considered for future work.

**4.3.2 Application to buckling prediction.** Results in Section 4.2.1 showed that the forecast was accurate for steady profile A. The forecast accuracy obtained for steady increase in force profiles is important to keep track of axial force increase alerts that occur before a needle

buckling is detected. Similarly, the forecast results for rapid force profiles (B,C) are also important for needle buckling prediction. Since initial rapid decrease was captured in the reference force patterns and continuous decrease in one direction was not possible owing to needle-tissue friction, the forecast at C was accurate. Essentially, when some increase in the axial force is alerted and captured as reference forces, a forecast for the axial force is obtained similar to C. However, force profile at B resembles that of a fluctuation in the force data with rapid increase and immediate decrease. Since the duration to capture this fluctuation in the reference force trends was small, the forecast accuracy at B was slightly higher. This result is important for predicting buckling events for duty-cycled insertions, since the corresponding force data consist of fluctuations similar to B. From Fig 7(e), the RMSE for profiles at (A,C) was lowest for $\hat{\Phi}_N(k_o) = 0.5$. Since the RMSE at B was not very high for $(\hat{\Phi}_N(k_o) = 0.5$ and fluctuations occur for low duty-cycles [69], the values $(\hat{\Phi}_N(k_o) = 0.5, \lambda = 0.1)$ would be used to develop the needle buckling prediction algorithm discussed in the next section.

## 5 Needle buckling prediction

A schematic overview of the needle buckling prediction and detection algorithm is shown in Fig 8. When a flexible needle steers into a tissue, axial force increases gradually due to frictional forces exerted by the tissue on the needle [19]. When the needle encounters a stiffer tissue or a hard membrane (say at time index $k = k_{fi}$), a rapid increase in the axial force occurs before the needle buckles within the tissue. On further insertion (at $k = k_b > k_{fi}$), the needle starts buckling within the tissue. However, to avoid this buckling at $k_b$, a prediction of the buckling should be made at $k_{fi}$ as soon as the force increase alert is detected. Using the proposed CFDL-MFP method, forecasts for the force data are obtained at $k_{fi}$ for a prediction horizon $N$. Using the $N$-

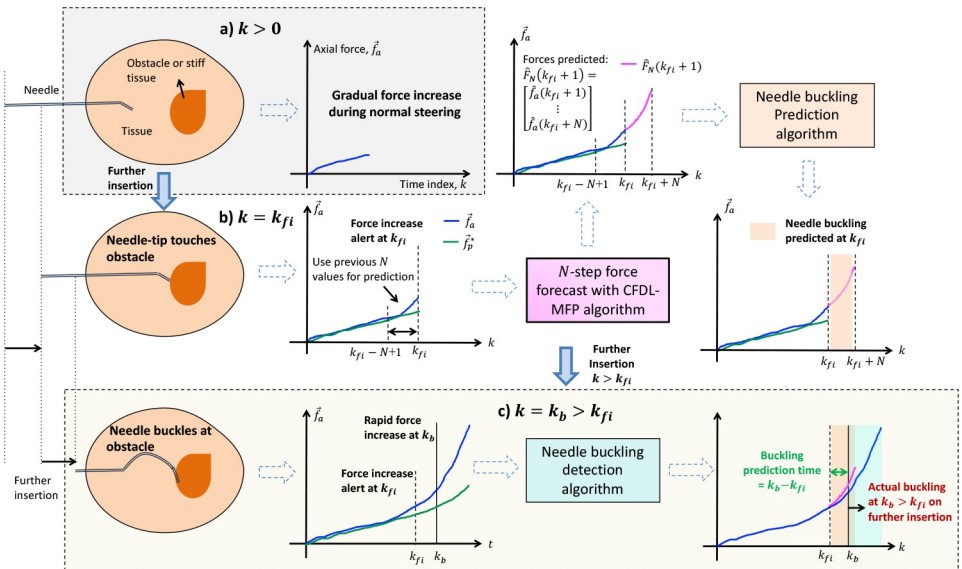

**Fig 8. Overview of needle buckling prediction and detection during insertions into a tissue.** a) Axial force increase is gradual under normal steering conditions. b) Rapid increase in force detected when needle-tip touches obstacle at time $k_{fi}$. $N$-step forecasts for force data produced by CFDL-MFP and buckling prediction algorithm uses forecasted forces to predict needle buckling at $k_{fi}$. c) On further insertion, needle buckles at the obstacle at time $k_b$ which is detected by the previously developed buckling detection algorithm. Buckling prediction time is calculated as $k_b - k_{fi}$, with positive value indicating early prediction. Solid blue arrows represent different stages of insertion, while dotted arrows represent sequential steps within the same time state $k$.

step forecasts and modifying the existing buckling *detection* algorithm [21], a *prediction* algorithm is developed to predict a potential needle buckling at $k_{fi}$. How soon a prediction is performed before a buckling actually occurs is determined by the prediction time $(k_b - k_{fi})$. Note that prediction time is relevant only for evaluating the prediction algorithm. For real-time implementation, buckling of the needle should be avoided through appropriate preventive subroutines as soon as a buckling is predicted at $k_{fi}$.

## 5.1 Prediction methods

The detection steps for axial force increase and needle buckling events are presented with respect to the CFDL-MFP forecasted forces. The complete details of the buckling detection algorithm including the procedure for generating optimal mean force, $\vec{f}_p^*$ and the selection of the sigmoid detection metrics, $M$ can be found in [21]. As a first step, the difference between $\vec{f}_a(k)$ and $\vec{f}_p^*(k)$ is normalized using a sigmoid function, $g$ to obtain $M$:

$$
\begin{aligned}
z^*(k) \quad &= \vec{f}_a(k) - \vec{f}_p^*(k) \\[2mm]
g(z^*(k)) \quad &= \frac{1}{1 + e^{-z^*(k)}} \\[2mm]
\text{with metrics, } M \quad &= \begin{cases} m_1 = g(1) = 0.73 \\[2mm] m_2 = g(2) = 0.88 \end{cases}
\end{aligned}
\tag{20}
$$

When the normalized difference, $g(z^*(k))$ exceeds the threshold $(m_1)$, and if the velocity of the needle-tip drops below the commanded insertion velocity, $v$, a force increase alert at time index $k$ is detected as follows:

$$
\text{If } \left( g(z_k^*) \in (m_1, m_2) \text{ and } 0 < \{ \frac{|\Delta l(k)|}{\Delta \tau(k)} < v \right)
\tag{21}
$$

$$
\Rightarrow \text{ Force increase alert}
$$

where $l$ is the euclidean norm of 3D needle-tip position and $\Delta l = l(k) - l(k-1)$ is the position change with respect to insertion time $\tau(k)$. As soon as the force increase is alerted, CFDL-MFP provides $N$-step force forecasts, $\hat{F}_N(k+1)$. The difference between $\hat{F}_N(k+1) \in \mathbb{R}^N$ and $\vec{f}_p^*(k)$ is normalized to predict a potential buckling as follows:

$$
\hat{Z}_N^*(k+1) \quad = \hat{F}_N(k+1) - \left[ \vec{f}_p^*(k), .., \vec{f}_p^*(k) \right]^T \in \mathbb{R}^N
$$

$$
\hat{G}_N(k+1) \quad = \begin{bmatrix} \hat{g}(k+1) \\ \hat{g}(k+2) \\ .. \\ \hat{g}(k+N) \end{bmatrix} = \frac{\vec{\mathbf{1}}}{\vec{\mathbf{1}} + \mathbf{e}^{-\hat{Z}_N^*(k+1)}}
\tag{22}
$$

$$
\text{If } [\hat{g}(k+i) \in (m_1, m_2) \text{ or } \hat{g}(k+i) > m_2]; \quad i = 1, 2, .., N
$$

$$
\Rightarrow \text{ Needle buckling predicted}
$$

Note that only rapid increase of force forecast patterns were used to predict needle buckling.

Prediction of the needle-tip position was ignored since the rate of change in the position was already smaller than the insertion velocity during axial force increase alert. With further increase in the force, the needle would either stop steering causing a buckling or puncture through the tissue.

**Algorithm 3** NEEDLE BUCKLING PREDICTION

**Require:** Axial force ($\vec{f}_a$), normalized needle-tip position ($l$), and needle insertion velocity ($v$), and total insertion time ($K_s$)

```
1:  Initialize k
2:  while k < K_s do
3:      f⃗_p*(k) ← optimalMean (f⃗_a(k))
4:      g(k) ← sigmoid (f⃗_a(k),f⃗_p*(k))
5:      if k = N then
6:          k_o ← k
7:          Initialize CFDL-MFP parameters
8:      end if
9:      if m_1 < g(k) < m_2 and 0 < |Δl(k)| < v then
10:         k_fi ← k
11:         Obtain CFDL-MFP forecasts, F̂_N(k+1)
12:         Ĝ_N(k+1) ← sigmoid (F̂_N(k+1),f⃗_p*(k))
13:         Initialize i ← 1
14:         while i < N do
15:             if ĝ(k_fi + i) > m_1 then
16:                 Buckling predicted at k_fi
17:             end if
18:         end while
19:     end if
20: end while
```

## 5.2 A complete prediction algorithm

A complete iterative algorithm integrating CFDL-MFP forecast and buckling prediction methods is presented in Algorithm 3. The inputs to the algorithm are the axial force data from a force sensor, $\vec{f}_a(k)$, normalized needle-tip position from a position sensor, $l(k)$, commanded insertion velocity, $v$, and total insertion time $K_s$. When the insertion begins from k = 0, the optimal mean forces are generated from $\vec{f}_a(k)$ at every time $k > 0$, to continuously check for force increase alerts through sigmoid transformation, steps (3-4). After ignoring the first $N$ values, the CFDL-MFP parameters are initialized according to the procedure described in Algorithm 1, (steps 4-6). When a force increase alert is detected at $k = k_{fi}$, N-step forecasts are obtained at $k_{fi}$, steps (9-11). Each forecasted value of the force within $N$ is checked for a rapid increase. If the rapid increase is detected, then a buckling is predicted at $k_{fi}$, (steps 12-18). This process continues till the end of the simulation at $K_s$. Therefore, the prediction algorithm can be implemented in real-time since both force increase detections and forecasts are performed during each iteration. Note that force forecasting is performed only when a rapid force increase is alerted. Prediction is not needed if a needle buckling is detected directly since appropriate control action would be implemented immediately. Hence, force forecasting and buckling prediction are performed only when a force increase is alerted.

## 6 Experimental results and discussion

In this section, we present methods to validate the buckling prediction algorithm through insertions in three different tissue environments; 1) gelatin tissue with embedded obstacle ($T_1$), 2) two-layered gelatin tissue ($T_2$), and 3) *ex-vivo* tissue ($T_3$) and evaluate the performance

in terms of prediction time, false predictions of buckling and false negatives (failed predictions). The results for all needle insertions are explained in the subsequent sections along with the statistical analysis of the performance metrics.

## 6.1 Methods and protocol

Needle insertion data from [21] were used to evaluate the buckling *prediction* algorithm with respect to the buckling events previously detected by our *detection* algorithm. The corresponding tissue was made of a uniform gelatin with a 3D printed plastic rectangular barrier embedded at a distance of 11cm from the shortest edge as shown in [21]. As short-hand notation, let this tissue with the known embedded obstacle be $T_1$. Next, to evaluate the effects of varying tissue stiffness on the prediction algorithm, the insertion data from a two-layered gelatin tissue were used where the less dense gelatin was placed at 8.5cm from the shortest edge [70]. Let this two-layered tissue be denoted as $T_2$. Finally, the prediction algorithm is tested for robustness and generalizability through insertions in *ex vivo* liver under two boundary conditions; a) unconstrained and b) constrained. Let this tissue be denoted as $T_3$. For manual verification, insertions in $T_3$ were recorded on the 9-in XRII OEC series 9600 fluoroscope as shown in [21].

The insertion data for each gelatin tissue consist of 75 needle insertions repeated 5 times for each of the three insertion velocities ($v$ = 1,2.5,5)mm/s and five duty-cycles (DC = 0,25,50,75,100)% with bidirectional duty-cycle spinning control implemented [69]. Six insertions from unconstrained $T_3$ and nine insertions from constrained $T_3$ were used, totaling to 15 needle insertions. Each *ex vivo* insertion consisted of only $v$ = 2.5mm/s and DC = 0% to avoid any coring effect of the DC rotations on the tissue structure [71]. The force data and needle-tip position were collected at a sampling rate of 20ms (50Hz) from a 6-axis force-torque sensor (Nano-17, ATI Inc.), and an electromagnetic (EM) tracker (TrakStar, NDI, Inc.), respectively. The steerable needle was made from a hollow nitinol tube with dimensions of 250mm length, 0.8mm OD, and 0.6mm ID, with a pre-bent tip of length 4mm long and angle 30 [69].

The prediction algorithms were implemented in MATLAB® R2017a. To imitate real-time prediction, the data was used as a sequential input to Algorithm 1 and Algorithm 3 where each value was available only at current time $k$. When $k > 0$, the prediction horizon $N$ for $t$ = 1.5s was obtained according to Eq (16). Since the sampling interval is 0.02s, $N$ = 71. For all insertions, force forecasts with CFDL-MFP were obtained for $\lambda$ = 0.1 and initial PPD = 0.5. The total computation time required to forecast force data and predict needle buckling was captured through steps (9-19) of Algorithm 3.

At the end of the simulation, the following performance metrics of the buckling prediction algorithm were evaluated: 1) Buckling prediction time as the difference between the predicted time stamp and the detected buckling timestamp, 2) True predictions (TP) resulting in exact prediction before a buckling detection, 3) False predictions of buckling (FP) due to force increase alert but no buckling detections, and 4) False negatives (FN) when actual buckling was detected but prediction failed. Finally, the effects of varying duty-cycle, insertion velocities and tissue environments on the metrics were evaluated through the standard statistical Kruskal-Wallis analysis.

## 6.2 Results in gelatin tissue with obstacle

The plots of insertion force and the needle insertion path obtained from normalized needle-tip position tracked by the electromagnetic sensor for three example needle insertions in $T_1$ are shown in Fig 9(a)–9(c). As the needle tip encountered the obstacle at 11cm, an axial force increase was alerted at 11.03cm for a DC = 100% insertion. At this time, forecasts for the force

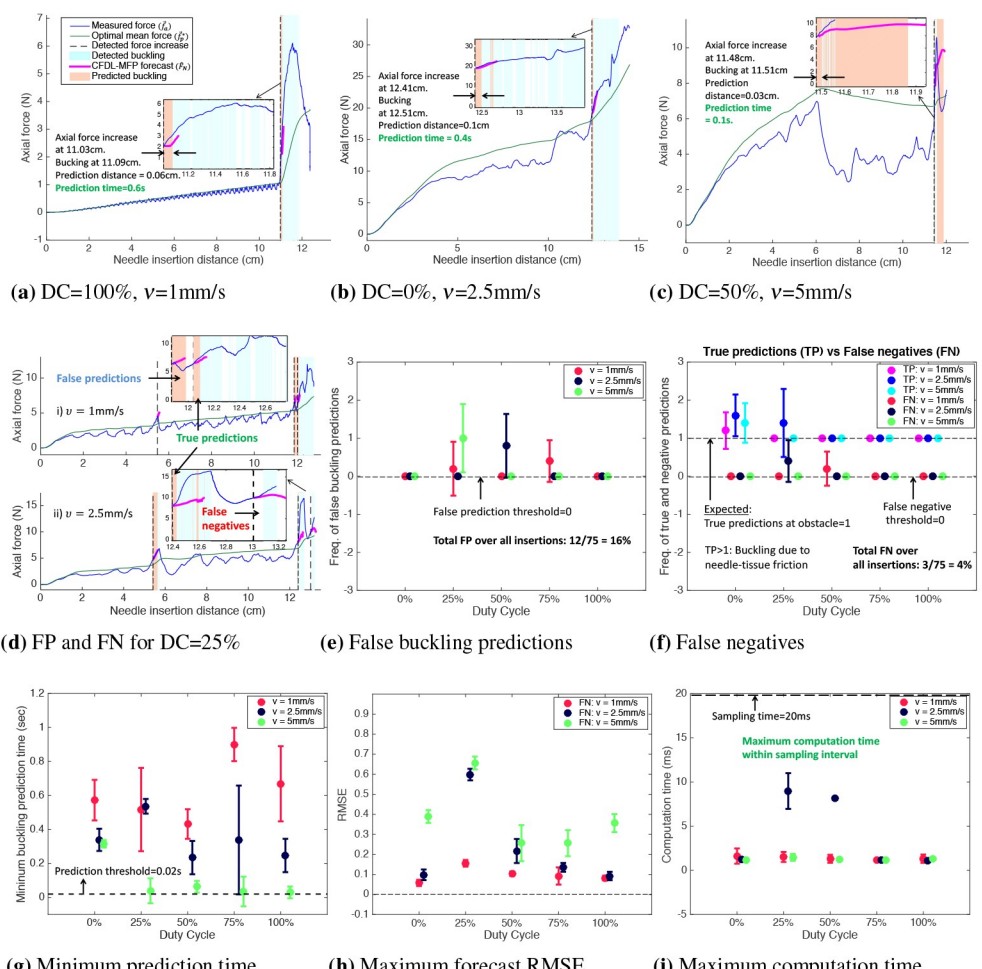

**(a)** DC=100%, *v*=1mm/s  **(b)** DC=0%, *v*=2.5mm/s  **(c)** DC=50%, *v*=5mm/s

**(d)** FP and FN for DC=25%  **(e)** False buckling predictions  **(f)** False negatives

**(g)** Minimum prediction time  **(h)** Maximum forecast RMSE  **(i)** Maximum computation time

**Fig 9. Buckling prediction results in $T_1$ with obstacle at 11cm from insertion point.** a) For $v$ = 1mm/s, buckling was predicted 0.06cm (0.6s) before detection at 11.09cm, with FP = 0 and FN = 0. b) Prediction time was 0.4s for $v$ = 2.5mm/s with FP = 0 and FN = 0. c) Prediction time was 0.1s for $v$ = 5mm/s, with FP = 0 and FN = 0. d) Plot showing FP and FN for DC = 25% insertion. e) Error bar plot showing FP > 0 for DC = (25,50,75)% insertions. f) Error bar plot showing FN > 0 for DC = (25,50)%. TP > 1 for DC = (0,25)% due to needle-tissue friction. g) Error bar plot of minimum buckling prediction time for each $v$ and DC. h) Error bar of RMSE which is high for DC = 25% due to fluctuations in the duty-cycled insertion force data. i) Total computation time (forecast+buckling prediction) for each insertion velocity and DC, which is less the 0.02s sampling interval.

were obtained from which buckling was predicted, as shown in Fig 9(a). Since the buckling detection timestamp recorded was 11.09cm, a buckling was predicted 0.06cm (0.6s for $v$ = 1mm/s) ahead before detection. With DC = 100% insertions, the needle insertion path was equal to the distance of the obstacle. Due to inherent kinematics, the insertion path was greater than 11cm for lower duty-cycles due to which a force increase was alerted at 12.41cm for DC = 0% insertion and a buckling was predicted 0.1cm (0.4s for $v$ = 2.5mm/s) ahead before detection. Similarly, the prediction time for an insertion with DC = 50% and $v$ = 5mm/s was 0.1s. In these cases, TP = 1 for a successful prediction whenever a buckling event occurred.

There were cases when force increase alerts did not lead to a buckling, but a buckling was still predicted. An example of this false prediction is shown in Fig 9(d) for DC = 25% and $v$ = 1mm/s insertion. A force increase was alerted at 11.8cm due to a fluctuation in the force data.

Since the point of force increase alert was at the rising trend of the fluctuation, the forecast trend also increased following the previous force patterns. The forecast accuracy was slightly low due to the immediate drop in the force data (similar to profile B from Section 4.2.1), and thus, FP = 1 for this insertion. From the error bar in Fig 9(e), FP > 0 for DC = (25,50,75)% due to frequent fluctuations in the force data caused by the bidirectional spinning method [72].

Next, false negatives occurred when a prediction failed and a buckling event was detected, as shown in Fig 9(d) for $v$ = 2.5mm/s insertion. This is due to the periodic forecasts which resulted from the periodic patterns of the duty-cycled insertion force data. Since the trend of the forecast (magenta plot) dropped, the forecast accuracy was low and thus, FN = 1 at 13cm. From the error bar plot in Fig 9(f), FN > 0 for DC = (25,50)% mainly due to the intrinsic periodic patterns. Both FP and FN occurred due to inaccuracies in the forecast and the percentage of their occurrence across 75 insertions were 16% and 4%, respectively. Furthermore, additional buckling occurred for lower duty-cycles since the deflection force was high causing the needle to bend along the insertion path [73]. The proposed algorithms successfully predicted this buckling resulting in TP>1 for DC = 0% and 25% insertions.

The buckling prediction time was high for lower insertion velocities, as evident in Fig 9(g). The prediction threshold was assigned as the sampling interval (0.02s) so that a buckling event was predicted before the next data arrived. If the incoming data lead to a buckling detection, the prediction time would have been close to 0.02s which is too less a time for the surgeon to implement a preventive action. Thus, an ideal prediction time is expected to be greater than the threshold with higher positive values indicating sooner predictions. The minimum buckling prediction time for all insertions with $v$ = 5mm/s was 0.03s, which was still slightly higher than the sampling interval. The effect of RMSE on duty-cycle and insertion velocity shown in Fig 9(h) are discussed later in the statistical analysis section. Finally, the maximum computation time captured during both force forecasting and prediction of needle buckling, was less than 10ms which is well within the sampling interval of 20ms as shown in Fig 9(i). The computation time for each forecast and buckling prediction does not change and will therefore, not be used in statistical analysis.

## 6.3 Results in two-layered gelatin tissue

Results in $T_2$ are analyzed for the frequency of the occurrence of FP, FN and TP due to needle-tissue friction. For example, FP occurred for DC = 75% and $v$ = (1, 2.5)mm/s due to a force increase alert at 7.6cm and 8.6cm, respectively as shown in Fig 10(a). The forecast accuracy

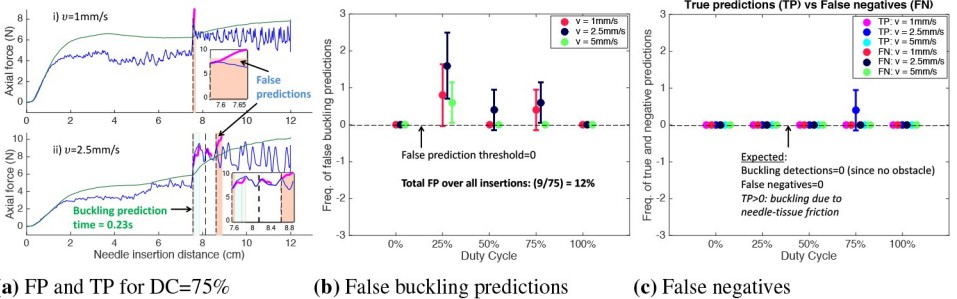

**(a)** FP and TP for DC=75%    **(b)** False buckling predictions    **(c)** False negatives

**Fig 10. Buckling prediction results in $T_2$ with less dense layer at 8.5cm from insertion point.** a) Plot showing FP for DC = 75% and $v$ = (1,2.5)mm/s. Buckling detected at 7.83cm for $v$ = 2.5mm/s due to high friction force of denser gelatin, resulting in TP = 1 for successful prediction with prediction time = 0.23s. b) Error bar plot showing FP > 0 for DC = (25,50,75)%. c) TP = 2 for two repeated insertions of DC = 75% and $v$ = 2.5mm/s, with FN = 0.

was low since the points of force increase alert were at the intersection of the rapidly changing forces, similar to profiles (E,F) from Section 4.2.1. Since, the trends of the forecasts increased, a false buckling was predicted resulting in FP = 1 at these points. However, a buckling was detected at 7.83cm due to the high friction force of the dense gelatin layer. The prediction algorithm successfully predicted this buckling 0.057cm (0.23s) before 7.83cm, resulting in TP = 1. An additional force increase alert was detected at 8.15cm, and the CFDL-MFP forecast at this point followed the actual force pattern resulting in high accuracy. Consequently, FP = 0 and FN = 0 which is trivial since TP = 0. From the error bar plot in Fig 10(b), FP > 0 was common for DC = (25,50,75)%, with the percentage of occurrence as 12% across 75 insertions in $T_2$. Two buckling events were detected for DC = 75% and $v$ = 2.5mm/s insertion, which might have due to tissue inhomogeneity. The prediction of these events was successful, resulting in TP = 2 as shown in Fig 10(c). Finally, there were no false negatives for any insertion.

## 6.4 Results in *ex-vivo* tissue

The plots for example insertions in $T_3$ for unconstrained and constrained boundary conditions along with the corresponding fluoroscopic motion images are shown in Fig 11. The motion

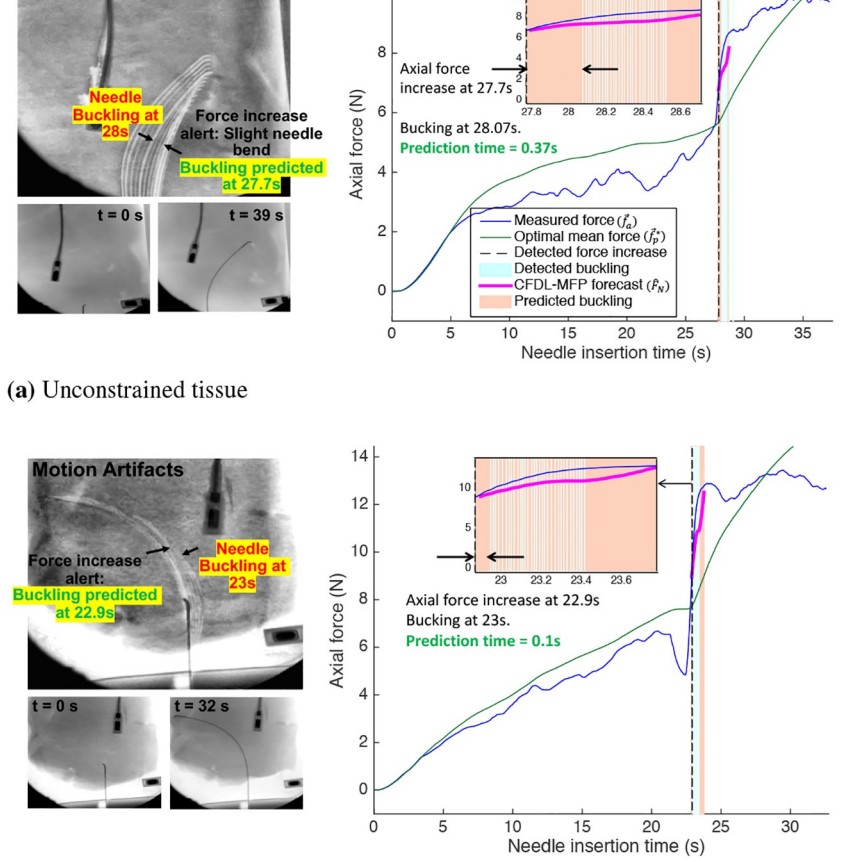

**(a)** Unconstrained tissue

**(b)** Constrained tissue

**Fig 11. Buckling prediction results for insertions in *ex vivo* tissue under unconstrained (a) and constrained (b) boundary conditions.** a) Axial force increase alert at 27.7s. Slight needle bend was visible from motion image. Needle buckling was predicted 0.37s before needle completely buckled at 28.07s. b) Similarly, needle buckling was predicted 0.1s before needle buckled at 23s.

**Table 4. Needle buckling predictions in *Ex vivo* tissue.**

| Parameter | Boundary Condition I | Boundary Condition II |
|---|---|---|
| Total insertions | 6 | 9 |
| No. of buckling detections (Ground truth) | 2 | 4 |
| Mean prediction time (sec) | 0.348 (SD = 0.033) | 0.165 (SD = 0.076) |
| Mean prediction time sooner than human detections (sec) | 1.67 (SD = 1.06) | 2.39 (SD = 1.20) |

images were captured from the live fluoroscopic videos of needle insertions at a frame rate of 30fps. For the unconstrained tissue, a slight bend in the needle shaft was visible from the motion image when an axial force increase was alerted at 27.7s, shown in Fig 11(a). At this time, forecast for axial force was performed (magenta plot) and a buckling was predicted (peach orange patch). The prediction was successful with a prediction time of 0.37s, since the needle buckled later at 28s. Similarly, the prediction time for the constrained tissue was 0.1s sooner than the buckling detection at 23s, as shown in Fig 11(b). A summary of the buckling predictions for all insertions in the biological tissue is given in Table 4. The mean buckling prediction time was 0.348±0.033s for the unconstrained tissue which was almost double the mean prediction time of 0.165±0.076s for the constrained tissue. During needle insertions, the unconstrained tissue displaced more frequently causing the needle shaft to bend laterally [70]. Due to this bend, axial force increased slightly, providing an alert even before the needle tip encountered an obstacle. Since this bend eventually lead to a buckling, the prediction time was high. However, the prediction time was lower for insertions in the constrained tissue. Any deformations within the constrained tissue would have caused the force to increase rapidly. As a result of this rapid increase, when buckling occurred with arrival of the next data points, the prediction time was low but still higher than the sampling interval for successful prediction. On average, the algorithm predicted buckling events sooner than the human volunteers by 1.67s and 2.39s for the two boundary conditions, respectively. Finally, no false predictions and false negatives were observed for any insertion.

## 6.5 Statistical analysis and discussion

Results from the statistical Kruskal-Wallis analysis on the needle insertions across all duty-cycles, insertion velocities and tissue types (166 insertions) are shown in Table 5. A significance level $p = 0.001$ was chosen.

**6.5.1 Effects of insertion velocity.** The buckling prediction time reduced as $v$ increased. Since larger distance is covered by the needle-tip in every second for high $v$, the needle tip encounters an obstacle faster and subsequently, rapid increase in the force is alerted with the next incoming data point. In such cases, the prediction time was close to the sampling interval of 0.02s, see Fig 9(g). Moreover, the number of discrete data observations reduces as $v$ increases for a fixed sampling interval and total insertion time. Therefore, the time series like

**Table 5. Statistical analysis—Buckling prediction.**

| Parameter | Insertion Velocity (mm/s) | | Duty-cycle (%) | | Tissue type | |
|---|---|---|---|---|---|---|
| (Mean values) | $p$ | Significant Groups | $p$ | Significant Groups | $p$ | Significant Groups |
| Buckling prediction time | <0.001 | $1 > 2.5 > 5$ | 0.022 | - | 0.8032 | - |
| False predictions | 0.5485 | - | <0.001 | $25 > (0,50,75,100)$ | < 0.001 | $(T_1, T_2) > T_3$ |
| False negatives | <0.001 | $(1,2.5) > 5$ | <0.001 | $(25,50) > (0,75,100)$ | < 0.001 | $T_1 > (T_2, T_3)$ |
| CFDL-MFP forecast RMSE | <0.001 | $5 > 2.5 > 1$ | <0.001 | $25 > (0,50,75,100)$ | < 0.001 | $T_1 > (T_2, T_3)$ |

force data become short which generally have low forecast accuracy [54]. As a result, RMSE increased with $v$, see Fig 9(h). Since false negatives were observed only for ($v$ = 1, 2.5)mm/s insertions in $T_1$, there was significant effect ($p < 0.001$) with FN = 0 for $v$ = 5mm/s and FN > 0 for $v$ = (1, 2.5)mm/s, (see Fig 9). However, it is difficult to determine the exact effects of $v$ with the small sample size of FN = 3 out of 166 insertions. Finally, there was no significant effect on FP ($p = 0.548$).

**6.5.2 Effects of duty-cycle, DC.** DC had significant effect on FP and FN due to the reasons mentioned in Section 7.1. Since FP and FN are most likely caused by incorrect forecasts, RMSE had a direct relation with FP and FN. Therefore, RMSE was significantly high for DC = 25% insertions. However, there was no significant effect on the buckling prediction time ($p = 0.022$).

**6.5.3 Effects of tissue type.** To evaluate the effects of tissue type more accurately, the metrics are averaged across all insertions with only DC = 0% and $v$ = 2.5mm/s. By observation, the mean prediction time for $T_1$, $T_2$ and $T_3$ was 0.3s, 0.23s and 0.25s, respectively for same DC and $v$. Therefore, tissue type had no significant effect on the prediction time ($p = 0.8032$). Next, forecast RMSE was most significant for insertions in $T_1$ due to non-zero FP and FN. Finally, both FP = 0 and FN = 0 for all insertions with DC = 0%.

**6.5.4 Discussion.** Results show that the prediction algorithm can be implemented accurately for insertions in a variety of tissue environments. The low percentage of false positives and false negatives show that the prediction algorithm can be implemented to alert a surgeon of a potential buckling event accurately without many false alarms. To provide surgeons enough time to implement preventive actions for the needle before buckling, the prediction time should be significantly high. Even with the advantage of high prediction times, low insertion velocities can cause larger tissue deformations [74]. Thus, an optimal insertion velocity should be determined such that prediction time is high and tissue deformation is low. If very low frequency of false predictions and false negatives is desired for any duty-cycled insertion, then the forecast RMSE will need to be improved in future work. As we have seen in Section 4.2.3, optimal forecasts need not necessarily produce accurate forecasts. Depending on the force profile at which forecasting is required, the CFDL-MFP parameters can be tuned for better accuracy. For example, if forecast is required at the intersection of rapidly changing profiles, then underestimated or linear forecasts can provide better accuracy (recall Fig 7(b) for profile F), resulting in smaller FP and FN. Therefore, an optimized model to determine optimal insertion inputs and adaptive forecasting parameters are required for better prediction accuracy.

# 7 Conclusion

We presented a novel time series forecasting method by modifying the existing model-free adaptive control algorithm [48] which enables fast computational times and does not require model-fitting or training data to perform single-step or multi-step predictions. Our CFDL-MFP forecasting model is **11%** more accurate than the benchmark statistical ARIMA method. The computation time of our proposed forecasting method is in the order of **4ms** for a 20ms data sampling rate, which enables real-time computation and future integration with robotic control systems. The stability, generalizability and robustness of the CFDL-MFP algorithm was proved through statistical analysis on a variety of force insertion profiles.

The secondary goal of this paper is to predict needle buckling events before they occur so that potential damage to the tissue and needle could be avoided. The buckling prediction algorithm was evaluated using steerable needle insertions into three types of tissue including artificial gelatin and *ex vivo* liver. The steerable needles were also controlled at various insertion

velocities and duty-cycles to control needle curvature. Overall, the false positive and false negative rates were low in artificial tissue (16% false positive and 4% false negative), and non-existent in *ex vivo* tissue.

One limitation of our proposed algorithm is that we can only distinguish when buckling events occur, but not necessarily the cause of those events (e.g., a stiffer portion of tissue vs. a rigid obstacle). In future work, we plan to integrate our CFDL-MFP buckling prediction algorithm with an existing steerable needle robot to determine the effectiveness of this technology in preventing needle buckling events in realistic clinical environments and evaluate the ability of the prediction algorithms to enable improved shared control between the human operator and steerable needle. Additionally, applying the CFDL-MFP algorithm to predict other important events in needle-based insertion such as tissue puncture or tissue displacements could be an interesting direction for future work. The proposed forecasting method could also be applied to other important problems in human-robot interaction research where real-time sensor data is critical for determining safe future robot control actions.

## Author Contributions

**Conceptualization:** Meenakshi Narayan.

**Formal analysis:** Meenakshi Narayan.

**Investigation:** Meenakshi Narayan, Ann Majewicz Fey.

**Methodology:** Meenakshi Narayan.

**Software:** Meenakshi Narayan.

**Supervision:** Ann Majewicz Fey.

**Writing – original draft:** Meenakshi Narayan.

**Writing – review & editing:** Ann Majewicz Fey.

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
