## [Decision Letter · Decision Letter 0]

9 Jan 2020

PONE-D-19-33507

Developing a novel force forecasting technique for early prediction of critical events in robotics

PLOS ONE

Dear Ms Narayan,

Thank you for submitting your manuscript to PLOS ONE. After careful consideration, we feel that it has merit but does not fully meet PLOS ONE’s publication criteria as it currently stands. Therefore, we invite you to submit a revised version of the manuscript that addresses the points raised during the review process.

We would appreciate receiving your revised manuscript by Feb 23 2020 11:59PM. To enhance the reproducibility of your results, we recommend that if applicable you deposit your laboratory protocols in protocols.io, where a protocol can be assigned its own identifier (DOI) such that it can be cited independently in the future. For instructions see: http://journals.plos.org/plosone/s/submission-guidelines#loc-laboratory-protocols

We look forward to receiving your revised manuscript.

Kind regards,

Huichan Zhao

Academic Editor

PLOS ONE

Journal Requirements:

2. Our internal editors have looked over your manuscript and determined that it is within the scope of our Open Soft Robotics Research Call for Papers. This collection of papers is headed by a team of Guest Editors for PLOS ONE: Guoying Gu (Shanghai Jiao Tong University), Aslan Miriyev (EMPA), Lucia Beccai (IIT), Matteo Cianchetti (Scuola Superiore Sant'Anna), Barbara Mazzolai (IIT) and Dana Damian (University of Sheffield).

The Collection will encompass a diverse range of research articles on soft robotics ranging from the development of future soft robots, durability, reliability, reproducibility and resilience in challenging environments. Additional information can be found on our announcement page: https://collections.plos.org/s/soft-robotics.

If you would like your manuscript to be considered for this collection, please let us know in your cover letter and we will ensure that your paper is treated as if you were responding to this call. If you would prefer to remove your manuscript from collection consideration, please specify this in the cover letter. Please note that inclusion in the call for papers will not result in a delay to publication and that the Guest Editors have final approval on what submissions are included.

3. Additionally, for reproducibility reasons we would recommend that you provide a supplementary file or a link to where you have deposited your MatLab code so that other researchers may reproduce the findings.

5. Thank you for stating the following in your Competing Interests section: "No"

6. We noted in your submission details that a portion of your manuscript may have been presented or published elsewhere.

"Fig 9 has been taken from our previously published papers to let the readers know that the same experimental setup and data were used for validations. Since the experimental protocol and results are different, we feel this is not dual publication.

A copy of our two prior published work are attached named as 'previous_work1.pdf and 'previous_work2.pdf' and uploaded along with the main manuscript for reference."

Please clarify whether this publication was peer-reviewed and formally published. If this work was previously peer-reviewed and published, in the cover letter please provide the reason that this work does not constitute dual publication and should be included in the current manuscript.

Reviewers' comments:

Reviewer's Responses to Questions

**Comments to the Author**

1. Is the manuscript technically sound, and do the data support the conclusions?

Reviewer #1: Yes

Reviewer #2: Yes

2. Has the statistical analysis been performed appropriately and rigorously? 

Reviewer #1: Yes

Reviewer #2: Yes

3. Have the authors made all data underlying the findings in their manuscript fully available?

Reviewer #1: Yes

Reviewer #2: Yes

4. Is the manuscript presented in an intelligible fashion and written in standard English?

Reviewer #1: Yes

Reviewer #2: Yes

5. Review Comments to the Author

Reviewer #1: This paper proposes a novel CFDL-MFP method to predict time series data, and develops a needle bucking prediction algorithm based on the predicted data. With the proposed method, the accuracy and the real-time performance of the critical event prediction are greatly improved. Corresponding simulations and experiments are implemented to evaluate the proposed method. This paper is worth publishing in PLOS ONE after minor revision.

A few suggestions to improve the quality of the paper:

1. The Organization of the paper should be adjusted. The method should be described first, and the experimental evaluation should be given following the proposed method. So the organization of the paper should be sections 3, 5, 4, 6,7 in turn, and what is more, section 6,7 should be merged into one section.

2. It would be better if the proposed CFDL-MFP needle prediction algorithm could be evaluated on real clinical data. If so, the paper will be more convincible and have a bigger impact.

3. What is the extensibility of the proposed method? Could the proposed prediction method also be applied for other event prediction problems?

Reviewer #2: In this paper, the authors presented an interesting and novel prediction method with good accuracy and low computational time, which make the method applicable in real time computations. The forecast algorithm is used to predict needle buckling by the axial force and the needle-tip position data.

The paper is organized very well, the literature review is very complete, and the prediction model is presented with details. The results show that the proposed method can predict the needle buckling even in reality.

Suggestion: The authors concluded that the prediction results did not depend on the tissue type. It seems that in the buckling prediction logic, the tissue stiffness has not been considered. I suggest the authors to mention it as an assumption (if necessary).

Question: It seems that the tip-position and the axial force have been used for buckling prediction. It is not clear for me how you used the force and position data together for prediction. Could you explain it?

6. PLOS authors have the option to publish the peer review history of their article (what does this mean?). If published, this will include your full peer review and any attached files.

Reviewer #1: No

Reviewer #2: Yes: Amin Lotfiani

---

## [Author Response · Author response to Decision Letter 0]

22 Jan 2020

Respond to Reviewers are attached as a separate PDF document uploaded along with the manuscript and other documents.

---

## [Decision Letter · Decision Letter 1]

20 Feb 2020

Developing a novel force forecasting technique for early prediction of critical events in robotics

PONE-D-19-33507R1

Dear Dr. Narayan,

We are pleased to inform you that your manuscript has been judged scientifically suitable for publication and will be formally accepted for publication once it complies with all outstanding technical requirements.

With kind regards,

Huichan Zhao

Academic Editor

PLOS ONE

Additional Editor Comments (optional):

The authors have addressed all questions the reviewers proposed and now the paper is ready for publication.

Reviewers' comments:

Reviewer's Responses to Questions

**Comments to the Author**

1. If the authors have adequately addressed your comments raised in a previous round of review and you feel that this manuscript is now acceptable for publication, you may indicate that here to bypass the “Comments to the Author” section, enter your conflict of interest statement in the “Confidential to Editor” section, and submit your "Accept" recommendation.

Reviewer #1: All comments have been addressed

Reviewer #2: All comments have been addressed

2. Is the manuscript technically sound, and do the data support the conclusions?

Reviewer #1: Yes

Reviewer #2: (No Response)

3. Has the statistical analysis been performed appropriately and rigorously? 

Reviewer #1: Yes

Reviewer #2: (No Response)

4. Have the authors made all data underlying the findings in their manuscript fully available?

Reviewer #1: Yes

Reviewer #2: (No Response)

5. Is the manuscript presented in an intelligible fashion and written in standard English?

Reviewer #1: Yes

Reviewer #2: (No Response)

6. Review Comments to the Author

Reviewer #1: The authors have responded to all questions I am concerned with, and I am satisfied with the revision. I think the revised manuscript is suitable for publication.

Reviewer #2: (No Response)

7. PLOS authors have the option to publish the peer review history of their article (what does this mean?). If published, this will include your full peer review and any attached files.

Reviewer #1: No

Reviewer #2: Yes: Amin Lotfiani

---

## [Editor Report · Acceptance letter]

24 Apr 2020

PONE-D-19-33507R1

Developing a novel force forecasting technique for early prediction of critical events in robotics

Dear Dr. Narayan:

I am pleased to inform you that your manuscript has been deemed suitable for publication in PLOS ONE. Congratulations! Your manuscript is now with our production department.

With kind regards,

on behalf of

Dr. Huichan Zhao

Academic Editor

PLOS ONE